# Assessing the contribution of ENSO and MJO to Australian dust activity based on satellite and ground-based observations

Yan Yu[1], Paul Ginoux[2]

[1] Atmospheric and Oceanic Sciences Program, Princeton University, Princeton, NJ 08540, USA
[2] NOAA Geophysical Fluid Dynamics Laboratory, Princeton, NJ 08540, USA

*Correspondence to*: Yan Yu (yanyu@princeton.edu)

**Abstract.** Despite Australian dust's critical role in the regional climate and surrounding marine ecosystems, the controlling factors of the spatio-temporal variations of Australian dust are not fully understood. Here we assess the connections between observed spatial-temporal variations of Australian dust with key modes of large-scale climate variability, namely the El Niño-Southern Oscillation (ENSO) and Madden-Julian Oscillation (MJO). Multiple dust observations from Aerosol Robotic Network (AERONET), weather stations, and satellite instruments, namely the Moderate Resolution Imaging Spectroradiometer (MODIS) and Multi-angle Imaging SpectroRadiometer (MISR) are examined. The assessed multiple dust observations consistently identify the natural and agricultural dust hotspots in Australia, including the Lake Eyre Basin, Lake Torrens Basin, Lake Frome Basin, Simpson Desert, Barwon-Darling Basin, Riverina, Barkly Tableland, and lee side of the Great Diving Range, as well as a country-wide, austral spring-to-summer peak in dust activity. Our regression analysis of observed dust optical depth (DOD) upon an ocean Niño index confirms previous model-based finding on the enhanced dust activity in southern and eastern Australia during the subsequent austral spring and summer dust season following the strengthening of austral wintertime El Niño. Our analysis further indicates the modulation of the ENSO-dust relationship with the MJO phases. During sequential MJO phases, the dust-active center moves from west to east associated with the eastward propagation of MJO, with maximum enhancement in dust activity at about 120˚E, 130˚E, and 140˚E corresponding to MJO phases 1-2, 3-4, and 5-6, respectively. MJO phases 3-6 are favorable for enhanced ENSO modulation of dust activity, especially the occurrence of extreme dust events, in southeastern Australia, currently hypothesized to be attributed to the interaction between MJO-induced anomalies in convection and wind and ENSO-induced anomalies in soil moisture and vegetation.

## 1 Introduction

Australia represents a major contributor of dust to the Southern Hemisphere (Tanaka and Chiba, 2006), influencing regional climate and the marine ecosystems of the surroundings ocean basins. The substantial aerosol loading to the atmosphere from dust storms in Australia exerts a direct effect on the radiation budget through the absorption and scattering of incoming shortwave radiation and the absorption and emission of outgoing longwave radiation (Choobari et al., 2013; Miller et al., 2004; Tegen and Lacis, 1996). Dust aerosols also produce an indirect effect by influencing the nucleation, microphysics, development, and optical properties of clouds, thereby altering rainfall patterns (DeMott et al., 2010). Dust transport and

deposition over ocean affects ocean biogeochemistry through changes to the iron supply (Gabric et al., 2010; Jickells et al., 2005). Australia's arid and semi-arid regions provide key supply of iron to the Southern Ocean and Antarctica by dust, during the present (Li et al., 2010), glacial (Lamy et al., 2014), and interglacial (Revel-Rolland et al., 2006) periods. Since most of the Southern Ocean is iron-limited (Sunda and Huntsman, 1997), the transport and deposition of Australian dust largely determine its productivity and carbon uptake (Boyd et al., 2004; Gabric et al., 2002). Therefore, deeper understanding of Australia's dust sources and drivers of variability in dust emission and transport from Australia will have broad implications on the regional and global climate.

Current understanding of Australian dust activity largely extends from interpretation of ground-based observations and satellite aerosol products which have not been thoroughly inter-compared over this region. Using a Dust Storm Index (DSI) derived from dust storm and visibility observations made at Australian Bureau of Meteorology (BoM) stations (McTainsh et al., 1998), O'Loingsigh et al (2014) assessed the spatial distribution of frequency and intensity of dust activity at 160 stations across Australia. Based on dust loading derived from the satellite aerosol optical depth (AOD), namely the Deep Blue algorithm applied to Moderate Resolution Imaging Spectroradiometer (MODIS) data from the polar-orbiting Terra and Aqua satellites, Ginoux et al. (2012) identified the Lake Eyre Basin as the leading natural dust source in Australia, consistent with previous ground-based (McTainsh, 1989) and satellite-based (Bullard et al., 2008; Prospero et al., 2002) identification of dust sources. Ginoux et al (2012) further identified agricultural dust sources in the Murray-Darling Basin in southeastern Australia, including the Victorian Big Desert, Riverina, and the Barwon-Darling Basin, consistent with an earlier satellite-based dust source identification (Prospero et al., 2002), model-based wind erodibility during dry years (Webb et al., 2006), but less consistent with dust storm frequency from weather stations (McTainsh, 1989; McTainsh et al., 1989, 1998, 2007; O'Loingsigh et al., 2014). Ginoux et al. (2012) attributed the potential inconsistency in dust source maps among datasets to the various temporal coverage of each dataset. The apparent inconsistency between satellite and ground-based spatial distribution of dust frequency and intensity could also be a result of the differential spatial coverage of these datasets. Therefore, it is critical to rigorously cross-validate these observations of dustiness in Australia.

Observations and General Circulation Models (GCMs) have shown substantial variability in the occurrence and intensity of dust emissions across Australia on the interannual to decadal time scales, primarily driven by Pacific sea-surface temperature (SST) fluctuations, particularly El Niño-Southern Oscillation (ENSO) events (Bullard and Mctainsh, 2003; Evans et al., 2016; Lamb et al., 2009; Risbey et al., 2009; Strong et al., 2011; Webb et al., 2006). Corresponding to the ENSO-induced rainfall anomalies, during El Niño conditions, there is increased wind erosion in central and southeastern Australia, while during La Niña years the sources are shifted to the southwestern regions (Webb et al., 2006). Based on the Geophysical Fluid Dynamics Laboratory (GFDL) climate model (CM3), Evans et al. (2016) further uncovered the role of climate-vegetation interactions in amplifying and persisting ENSO's modulation on dust emission in southeastern Australia. The modulation on dust emission and loads was simulated to further amplify the ENSO-related rainfall variability across eastern Australia (Rotstayn et al., 2011). On longer time scales, Lamb et al (2009) revealed a pronounced and consistent dust maximum during 1959 to 1973 and a much more dust-free period after 1977 across the central-eastern Australia, based on

the frequency of dust events reported at weather stations. This regime shift in Australian dustiness was attributed to wind field changes associated with oscillations in the Pacific climate system, including the latitudinal displacement of the South Pacific Convergence Zone, and SST changes of the Pacific Decadal Oscillation and North Pacific Oscillation (Lamb et al., 2009). From the paleoclimate perspective, the geochemical characteristics of Australian dust deposition in New Zealand over the last 8000 years have been used to identify corresponding dust sources, thereby inferring patterns of aridity in eastern Australia and climate variability associated with ENSO during the Holocene (Marx et al., 2009).

On the intra-to-inter-seasonal time scales, the variability in Australia's atmospheric and terrestrial states features the dominant signal from the Madden–Julian Oscillation (MJO) (Notaro, 2018; Risbey et al., 2009; Wheeler et al., 2009; Yu and Notaro, 2020); yet, the potential role of MJO on Australia's dust emission and transport remains understudied. The MJO is characterized by eastward-propagating, large-scale wave-like disturbances in the Tropics, particularly across the tropical Indo-Pacific region, with a typical cycle of 30-60 days (Madden and Julian, 1971, 1972). During an MJO event, anomalous convection acts as a tropical heat source emitting stationary Rossby waves that propagate into the extratropics and significantly modulate the extratropical circulation (Matthews et al., 2004; Seo and Son, 2012). Previous modeling and observational analyses have identified significant influence of MJO on rainfall and circulation across Australia (Marshall et al., 2013; Risbey et al., 2009; Wheeler et al., 2009). An observational study by Wheeler et al. (2009) identified rainfall responses across extratropical Australia to MJO-induced changes in vertical motion and meridional moisture transport during austral summer and other seasons, respectively. Marshall et al. (2013) uncovered that the observed probability of an upper decile heat event varies according to MJO phase and time of year, with the greatest impact of the MJO on extreme heat occurring over southern Australia in austral spring during phases 2–3. The convectively-active MJO phases 5-6 are also responsible for anomalous vegetation growth in the northern Australian monsoon region (Notaro, 2018), which further supports circulation changes over a broader region across the continent (Yu and Notaro, 2020).

Furthermore, MJO may interact with ENSO on the modulation of Australian dust emission. First, surface westerly anomalies introduced by the MJO can force downwelling oceanic Kelvin waves and influence ENSO evolution (Kessler et al., 1995; McPhaden and Taft, 1988), thereby triggering ENSO's modulation on Australian dust emission. Second, ENSO modulates MJO's propagation, resulting in differentiated spatial-temporal evolution of MJO (Wei and Ren, 2019) and its influence on Australia's regional climate. Furthermore, the convection and circulation anomalies introduced by MJO can enhance or weaken the basic response of regional climate to ENSO events, depending on the phase of MJO, as demonstrated by an observational study on extreme precipitation over northern South America (Shimizu et al., 2017). Despite MJO's critical influence on the regional climate, its direct or indirect role in modulating dust emission or concentration in Australia has, to our knowledge, never been explicitly investigated in either observations or models.

Motivated by the current knowledge gap in the large-scale climate drivers of Australian dust activity, the present study first assesses the multiple satellite and ground-based observations of dustiness in Australia, and then establishes the connection between the observed spatio-temporal variations in Australian dust activity with ENSO and MJO. Sections 2, 3, and 4 present the data and methods, results, and conclusions/discussion, respectively.

## 2 Data and Methods

### 2.1 DOD

DOD is a column-integrated extinction by mineral particles. The current study examines DOD during 2000-2019 from MODIS onboard the polar-orbiting Terra and Aqua satellites, the Multiangle Imaging SpectroRadiometer (MISR) instrument (Diner et al., 1998) on Terra, and the ground-based AErosol RObotic NETwork (AERONET) (Holben et al., 1998) sun photometers, with distinct retrieval algorithms.

#### 2.1.1 MODIS

Following Pu et al. (2020), daily DOD is retrieved from collection 6.1, level 2 MODIS Deep Blue aerosol products (Hsu et al., 2013; Sayer et al., 2013), including aerosol optical depth (AOD), single-scattering albedo ($\omega$), and the Ångström exponent ($\alpha$). All the daily variables are first interpolated to a 0.1° x 0.1° grid using the algorithm described by Ginoux et al. (2010). To account for dust's absorption of solar radiation and separate dust from scattering aerosols, such as sea salt, we require the single-scattering albedo at 470 nm to be less than 0.99 for the retrieval of DOD. Based on the size distribution of dust and to separate it from fine particles, DOD is retrieved as a continuous function of AOD and Ångström exponent:

$$DOD = AOD \times (0.98 - 0.5089\alpha + 0.051\alpha^2). \quad (1)$$

This retrieval of DOD is on the basis of Ångström exponent's sensitivity to particle size (Eck et al., 1999) and the previously established relationship between Ångström exponent and fine-mode AOD (Anderson et al., 2005). Details about the retrieval process and estimated errors are summarized by Pu and Ginoux (2018a). MODIS DOD products have been widely used for the identification and characterization of dust sources (Baddock et al., 2009, 2016; Ginoux et al., 2012), as well as examination of variations in regional and global dustiness (Pu et al., 2019, 2020; Pu and Ginoux, 2017, 2018a). Following the recommendation from Baddock et al. (2016) and previous applications of MODIS DOD (Pu et al., 2019, 2020; Pu and Ginoux, 2017, 2018a), here we use DOD with a low-quality flag of QA = 1, under the assumptions that 1) dust sources are better detected using DOD with a low-quality flag, and 2) retrieved aerosol products are poorly flagged over dust source regions.

#### 2.1.2 MISR

Benefiting from its multiangle observations, MISR data can be used to directly retrieve AOD and particle properties (Diner et al., 1998). In the current study, Version 23, Level 2 MISR 550-nm coarse-mode AOD and nonspherical AOD at 4.4-km resolution (Garay et al., 2020) are compared with MODIS DOD. The MISR nonspherical AOD fraction is often referred to as "fraction of total AOD due to dust", as dust is the primary nonspherical aerosol particle in the atmosphere, especially over desert regions such as those found in the arid and semiarid regions in Australia (Kalashnikova et al., 2005). The MISR nonspherical AOD has been used to examine variations in dustiness in North Africa and the Middle East (Yu et al., 2013, 2016, 2018, 2020). To be consistent with the MODIS data, MISR coarse-mode AOD is referred to as MISR DOD here,

while the nonspherical AOD is referred to as nsAOD in the current study. Similar to our use of MODIS DOD with a low-quality flag, here we analyze the raw MISR DOD and nsAOD retrieval without quality filtering. MISR DOD and nsAOD are also interpolated to a 0.1° x 0.1° grid using the algorithm described by Ginoux et al. (2010). Due to its relatively narrow swath of ~380 km, MISR samples the study region about every 10 days.

### 2.1.3 AERONET

The Version 3, level 2 (cloud screened and quality assured) AERONET coarse-mode AOD at 500 nm obtained from the 18 sun photometers across Australia (Giles et al., 2019) and retrieved by the Spectral Deconvolution Algorithm (SDA) (O'Neill et al., 2003) is analyzed here along with DOD from MODIS and MISR. To be consistent with the MODIS product, AERONET coarse-mode AOD is referred to as AERONET DOD in the current study. In the analysis of annual mean and seasonal cycle, AERONET DOD monthly data are first screened by removing those months with than five days of records. To calculate annual means, years with less than five months of records were removed. Collocated DOD from AERONET and satellite products are further compared. Here a "collocated observation" is identified when there is available MODIS or MISR DOD over the 0.1° grid covering the AERONET site within ± 0.5 hour of the corresponding AERONET site observation. At each AERONET site, one satellite observation is often associated with multiple AERONET observations in time. In this case, AERONET observations are temporally averaged, resulting in only one pair of collocated and averaged satellite-AERONET DOD observations for a given collocated incident at each AERONET site.

### 2.2 DSI from weather stations

The present study analyzes meteorological records of dust activity, based on nine weather codes that are related to dust events as defined by the World Meteorological Organization (WMO). The meteorological records are obtained from the National Climatic Data Center (NCDC) global and U.S. Integrated Surface hourly data set at 1489 weather stations in Australia. Following O'Loingsigh et al. (2014), the daily Dust Storm Index (DSI) at a specific station is a weighted sum of dust activity, calculated by:

$$DSI = 5 \times SDS + MDS + 0.05 \times LDE \quad (2)$$

Where severe dust storm (SDS) = 1 if a decreased (code 33), stable (code 34), or begun/increasing (code 35) severe dust storm with visibility < 200 m is reported at least once and 0 otherwise; moderate dust storm (MDS) = 1 if a decreased (code 30), stable (code 31), or begun or increasing (code 32) slight or moderate sand or dust storm with visibility < 1000 m is reported at least once and 0 otherwise; and local dust event (LDE) = 1 if raised dust or sand (code 07), well developed dust whirls (i.e. "dust devils", 08), or distant or past dust storm (code 09) is reported at least once or 0 otherwise. The credibility and temporal stability of DSI was evaluated in detail by O'Loingsigh et al. (2014).

### 2.3 Large-scale climate indices, environmental variables, and associated analysis

#### 2.3.1 Ocean Niño Index and regression analysis

To assess ENSO's modulation on Australian dustiness, an Ocean Niño Index (ONI) is analyzed. ONI is calculated as the three-month running mean of Extended Reconstructed Sea Surface Temperature, Version 5 (ERSSTv5) (Huang et al., 2017) SST anomalies in the Niño 3.4 region (5˚N-5˚S, 120˚-170˚W), based on centered 30-year base periods updated every five year (Climate Prediction Center, 2020). The influence of ENSO on DOD and DSI are quantified based on regression. To account for the non-Gaussian distribution of DOD and DSI, here the significance of regression coefficient is obtained through a Monte Carlo permutation test with 1,000 iterations, following Yu and Notaro (2020). In each iteration, the time series of DOD or DSI is randomly scrambled, leading to a random estimate of the regression coefficient on ONI. The probability distribution function (PDF) of the random regression coefficients is used to test if the regressions in the original, non-permutated data are statistically significant. In the current study, a significance level of 0.05 is used to indicate statistically significant results.

#### 2.3.2 Real-time multivariate MJO index and composite/regression analysis

To assess the potential influence of MJO and its interaction with ENSO on Australian dust activity, the real-time multivariate MJO index (RMM) (Wheeler and Hendon, 2004) is examined. RMM is derived as the principal components (PCs) corresponding to the leading two empirical orthogonal functions (EOF) of the combined fields of near-equatorially averaged 850-hPa zonal wind, 200-hPa zonal wind, and satellite-observed outgoing longwave radiation (OLR) data. Longer-time-scale variability resulting from ENSO and other interannual variations with periods longer than about 200 days is removed prior to the EOF analysis. The combination of PC1 (RMM1) and PC2 (RMM2) of RMM reflects the magnitude and phase of the MJO. When the amplitude is greater than 1, eight MJO phases are determined by the sign of RMM1 and RMM2. Phases 1 and 2 mark the time when the MJO's convective envelope is centered near the western Indian Ocean, and phases 5-6 mark the time when the envelope is near the northern Australia (Wheeler et al., 2009).

Composite analysis is conducted for DOD, frequency of extremely high DOD, and DSI during each of the consecutive two MJO phases (phases 1-2, 3-4, 5-6, and 7-8), compared with the mean DOD and DSI. A Monte Carlo bootstrap test with 1,000 iterations is used to determine the significance of anomalies in dustiness during various MJO phases. In each iteration, daily dustiness measures are randomly sampled with the same size as a particular group of MJO phases. These randomly sample dustiness measures are used to construct a PDF of sample mean dustiness and test if the mean dustiness during specific MJO phases is lower than the $2.5^{th}$ or higher than the $97.5^{th}$ percentile of the PDF. Further, regression of dustiness upon ONI is performed for each MJO phase group to evaluate potential role of MJO in modulating ENSO's influence on Australian dustiness.

#### 2.3.3 Other environmental variables

To examine the potential mechanisms underlying the modulation of ENSO and MJO on Australian dustiness, we assess the connection between these large-scale climate drivers and various environmental factors such as surface wind speed, precipitation, soil moisture, and leaf area index (LAI) across Australia. The data sources of these environmental variables are outlined in Table 1. Regression and composite analyses are applied to these environmental variables, similar with those applied to the dustiness observations. To account for the non-Gaussian distribution of these environmental variables, the statistical significance of the regression and composite signals are evaluated by the aforementioned non-parametric approaches.

## 3 Results

### 3.1 Comparison of multiple observations of dustiness

MODIS DOD from both Terra and Aqua and station-based DSI consistently identify the natural and agricultural dust hotspots in Australia, including the Lake Eyre Basin, Lake Torrens Basin, Lake Frome Basin, Simpson Desert, Barwon-Darling Basin, Riverina, Barkly Tableland, and the lee side of Great Dividing Range (Figure 1). The annual mean MODIS DOD reaches 0.2 over Lake Eyre, Lake Torrens, and Lake Frome, where over 30% of days observe a DOD exceeding 0.2, the 98$^{th}$ percentile of all MODIS DODs across Australia. MISR DOD is generally lower than MODIS DOD and exhibits minimal spatial variation. Moreover, MISR only captures the margin of Lake Eyre Basin and Barkly Tableland and shows relatively low dustiness over the Barwon-Darling Basin and mostly invalid retrievals over the Lake Torrens and Lake Frome Basins. The spatial distribution of mean DOD from AERONET is largely consistent with the satellite observations. The apparently high DOD from MODIS and AERONET over the coastal region is likely caused by the abundance of sea salt aerosol and its mixture with dust and biomass burning aerosols.

DOD from AERONET, MODIS, and MISR exhibit a generally consistent seasonal cycle in dust activity, which peaks in austral spring to summer, namely November, December, and January, across most of the country (Figures 2 and 3). In particular, the seasonal cycle in DOD is generally consistent between all satellite instruments and AERONET at most sites in Australia. The seasonal cycle in DOD and DSI are highly consistent in Birdville and Tinga Tingana, located near the dust hotspots in Simpson Desert and Lake Eyre Basin, respectively (Figure 3). The inconsistent seasonal cycle of satellite and AERONET DODs at Coleambally (34.8˚S, 146˚E) in Riverina (Figure 3o) is likely due to the short AERONET record covering only November, 2001 to February, 2003. The largest disagreement between satellite and station-based observations of dustiness occurs over the Barwon-Darling Basin and its northern downwind regions in eastern Australia, where satellite DOD indicates peak dustiness in November while the station-based DSI indicates peak dustiness in September. Over the Lake Eyre-Torrens-Frome Basin, the morning satellite observations, namely MODIS-Terra and MISR, display a summertime peak in dustiness, while the afternoon satellite observation MODIS-Aqua indicates a springtime peak. This

contrast between the seasonal cycles in morning and afternoon dustiness suggests a seasonally varying diurnal evolution of dust emission in south-central Australia.

Although the general comparison between collocated satellite and AERONET DODs exhibits high quality of satellite retrievals over the majority of Australia, MODIS-Terra, MODIS-Aqua, and MISR all underestimates high DOD (Figures 4-5). Very few MODIS DOD retrievals reach lower than 0.005, likely due to the numerical limits of retrieving algorithm. Furthermore, both MODIS and MISR underestimate high DOD, especially when AERONET DOD exceeds 0.1 (Figure 4d-f). This underestimation of high optical depth has been reported by previous global validations of total AOD from MODIS (Sayer et al., 2019; Wei et al., 2019) and MISR (Garay et al., 2020), as well as MODIS DOD (Pu and Ginoux, 2018b). The underestimation of high DOD potentially leads to the deteriorated temporal correlations between satellite and AERONET DODs over the dustiest region near the Lake Eyre Basin, compared with less dusty regions in Australia (Figure 5). Given the distinct retrieval algorithms involved in the satellite and AERONET DODs, the moderate but mostly significant temporal correlations between satellite and AERONET DODs (Figure 5) demonstrate the reliability of satellite DODs in representing coarse-mode aerosol loads.

According to the comparison between collocated MODIS DOD and MISR DOD and nsAOD, as well as satellite DOD versus station dust storm observations, coarse-mode AOD is an acceptable approximation of dustiness over the dust hotspots in central and southeastern Australia (Figures 6-7). While correlation between hundreds of MODIS and MISR DOD exceeds 0.4 across the majority of the country, even exceeding 0.6 over central-eastern Australia, the correlation between MODIS DOD and MISR nsAOD reaches 0.6 only near the major dust source regions, e.g. Lake Eyre-Torrens-Frome Basin, Simpson Desert, and northern downwind of Barwon-Darling Basin. Furthermore, higher MODIS DOD is observed on reported dusty days at most stations in Australia, especially during country-wide local dust and moderate dust events, as well as severe dust events in the south and east with MODIS-Aqua (Figure 7). Insufficient number of collocations between ground observations and MISR overpasses likely leads to the minimal signal in MISR DOD, as previously shown in the annual mean DOD (Figure 1). Overall, the general consistency between collocated MODIS DOD, MISR DOD and nsAOD, and station dust observations provides confidence in the credibility of these records in the representation of dustiness in Australia. Considering the temporal and spatial coverage of each dataset, only results from MODIS DOD, represented by the average between the morning (Terra) and afternoon (Aqua) overpasses, and station DSI are presented in the following section of climatic modulation on Australian dustiness.

### 3.2 Modulation from ENSO and MJO on Australian dustiness

According to regression analysis applied to multiple dust observation data sets and Niño3.4 at various antecedent time, austral wintertime El Niño supports enhanced dust activity in southern and eastern Australia during the subsequent austral summer dust season (Figure 8). An SST anomaly of +1˚C in the Niño 3.4 region during June to August (JJA) leads to an increase in daily mean DOD of about 0.05 over the lee side of the Great Dividing Range, 0.04 over the Barwon-Darling Basin, and 0.03 over Riverina during the subsequent December to February (DJF). The +1˚C warming in the Niño 3.4 region

during JJA causes an increase in the frequency of extreme DOD of about 5% over the lee side of the Great Dividing Range and 2% over Riverina, and an increase in DSI of about 1% over the Barwon-Darling Basin and Riverina during the subsequent DJF. The El Niño condition in the austral autumn and winter also leads to enhanced dust emissions across the Simpson Desert and Barwon-Darling Basin during the subsequent September-November (SON).

The lagged response in Australian dust activity to ENSO is attributed to ENSO's persistent and cumulative influence on the regional soil moisture and LAI (Figure 9). The El Niño-induced inhibited rainfall across the eastern and central Australia (e.g Risbey et al. 2009) leads to the depletion of soil moisture and a resulting reduction in vegetation cover, thereby favoring dust emission. In austral summer, the El Niño-induced reduction in vegetation cover across eastern Australia likely causes a reduction in surface roughness and strengthened surface wind that further enhances dust emission. The response magnitude in soil moisture and LAI in austral summer and spring peaks after 3-6 months of the ENSO signal, supporting the 3-6 months lag in the dustiness response to ENSO. The currently identified importance of vegetation in the modulation of ENSO on dust emission in Australia confirms the model-based finding about the role of climate-vegetation interactions in amplifying and persisting ENSO's modulation on dust emission in southeastern Australia by Evans et al. (2016).

According to the composite of DOD, frequency of extreme DOD, and station-based DSI, dust-active center moves from west to east associated with the eastward propagation of MJO, with maximum enhancement in dust activity at about 120˚E, 130˚E, and 140˚E corresponding to MJO phases 1-2, 3-4, and 5-6, respectively (Figure 10 and Table 2). During MJO phases 5-6, namely the convection-active phases for Australia, the increased surface wind speed over the majority of the continent, especially over the dust hotspots in the Lake Eyre-Torrens-Frome Basin and Riverina, appears responsible for the enhanced dustiness (Figure 11). During other MJO phases, the enhanced dustiness over the central and eastern Australian dust hotspots seem to be associated with anomalously wet conditions. Given that central-southern Australia generally receive less than 1 mm of rainfall on an average day, we hypothesize that over these arid or semi-arid regions, enhanced rainfall during the MJO phases 3-6 in austral spring and summer associated with enhanced convection and occurrence of thunderstorms support higher occurrence of haboob type of dust events. Several case studies have reported haboob dust events in the central and eastern Australia (McTainsh et al., 2005; Shao et al., 2007). Strong et al. (2011) found that about 24% of dust storms in the lower Lake Eyre Basin during 2005-2006 are associated with thunderstorms. Our alternative hypothesis relies on the supply of fine particles by occasional flooding from MJO-induced storms. For supply-limited and/or transport-limited dust sources such as those in southeastern Australia, lack of occasional storms under drier conditions usually leads to the failure of sediment replenishment, thereby leading to anomalously inactive dust emission (Arcusa et al., 2020; Bullard and Mctainsh, 2003).

ENSO's regulation of dust emission varies in magnitude by MJO phases, with MJO phases 3-6 favorable for enhanced ENSO regulation on dust activity, especially the occurrence of extreme dust events, in southeastern Australia (Figure 12 and Table 2). An SST anomaly of +1˚C in the Niño 3.4 region in austral winter is associated with an increased DOD by over 0.05, an increased frequency of extreme DOD by over 5%, and an increased DSI by 2% over the Barwon-

Darling Basin and Riverina during MJO phases 5-6 in austral spring and summer. MJO phases 3-4 features a moderately enhanced dustiness over the Lake Eyre-Torrens-Frome Basin in response to antecedent El Niño. We hypothesize that the enhanced response in dustiness across the southeastern Australia to ENSO during MJO phases 3-6 are attributed to the interplay between MJO-induced anomalies in convection, rainfall, and wind and the ENSO-induced anomalies in soil moisture and vegetation. While the dry soils and diminished vegetation caused by El Niño provide favorable conditions for dust emission (Figure 9), the active convections and strengthened surface wind during MJO phases 3-6 likely triggers more dust emission and extreme dust events across southeastern Australia through either haboob type of dust events or additional sediment supply by occasional flooding (Figure 11).

## 4 Discussion and Conclusions

The current study investigates the contribution of large-scale climate variability represented by ENSO and MJO to the modulation of Australian dust activity on the intra-seasonal to interannual time scales. Multiple sources of dustiness measurements, namely DOD from MODIS, MISR, and AERONET, nsAOD from MISR, and DSI from weather station, are inter-compared in terms of their annual mean, seasonal cycle, and day-to-day variations over a 20-year period from 2000-2019. These assessed dust observations consistently identify the natural and agricultural dust hotspots in Australia, including the Lake Eyre-Torrens-Frome Basin, Simpson Desert, Barwon-Darling Basin, Riverina, Barkly Tableland, and lee side the Great Diving Ranges, and a country-wide dust peak during austral spring-to-summer, confirming the previous ground-based (McTainsh, 1989) and satellite-based (Bullard et al., 2008; Ginoux et al., 2012; Prospero et al., 2002) identification of dust sources. Furthermore, the intercomparison between the multiple dust observations demonstrates the credibility of MODIS DOD - a widely analyzed satellite dust observation with optimal temporal and spatial coverage - over the arid to semi-arid regions in the central and southeastern Australia. Regression analysis of MODIS DOD upon Niño 3.4 SST confirms the previous model-based finding by Evans et al. (2016) on the enhanced dust activity in southern and eastern Australia during the subsequent austral summer dust season following the strengthening of austral wintertime El Niño. Composites of dustiness during sequential MJO phases demonstrates the propagation of dust-active center from west to east associated with the eastward movement of MJO, with maximum enhancement in dust activity at about 120˚E, 130˚E, and 140˚E corresponding to MJO phases 1-2, 3-4, and 5-6, respectively. Our analysis further indicates the modulation of the ENSO-dust relationship with the MJO phases; MJO phases 3-6 are favorable for amplifying ENSO's modulation on dust activity, especially the occurrence of extreme dust events in southeastern Australia.

Although the current study demonstrates the general reliability of MODIS DOD over the arid and semi-arid regions in Australia, uncertainties of this product should be noted. For example, the retrieval of MODIS DOD relies on the light-absorbing and coarse-mode nature of dust and is unable to distinguish between dust and the coarse-mode part of biomass burning aerosols (e.g. Noyes et al., 2020), leading to potential miss-representation of dust/smoke aerosols over the wildfire hotspots in northern Australia (Van Der Werf et al., 2017). Given the potential contamination from biomass burning

aerosols, our interpretation of the currently examined connection between dust and climatic drivers mainly focuses on the central and southeastern Australia. In addition, for haboob dust events which often occurs with the presence of convective clouds, MODIS and MISR algorithms are unlikely to perform aerosol retrievals. Overall, the optimal spatial and temporal coverage of MODIS aerosol products with over 20 years' record warrant its application for studying the spatio-temporal variations and environmental drivers of global aerosol loads.

The current analysis on the connection between environmental factors, such as LAI, soil moisture, wind, and precipitation, and ENSO and MJO leads to the hypothetical mechanisms underlying the identified modulation of ENSO and MJO on Australian dustiness. We hypothesize that the dry soils and diminished vegetation resulting from the El Niño-induced rainfall reduction provide favorable conditions for dust emission during the subsequent season; the enhanced convective activity and strengthened surface wind during MJO phases 3-6 likely triggers more dust emission and extreme dust events across southeastern Australia during the El Niño-associated dry years, thereby amplifying ENSO's modulation on dust emission. Under the hypothesized mechanism, we expect more pronounced MJO-enhancement of ENSO's modulation on dust following El Niño than La Niña conditions. One explanatory hypothesis for this relationship builds partly on the occurrence of haboob dust storms and its connection with MJO-induced anomalies in deep convection over the southeastern Australia. An alternative hypothesis relies on the supply of sediments by MJO-induced storms and their resulting occasional flooding. These alternatives motivate further evaluation of these hypothesized mechanisms underlying the modulation of ENSO-MJO on dust emission across Australia in an Earth System Model. In addition, the present study focuses on the natural drivers of Australian dust activity; while anthropogenic dust emission from land use change is a key contributor to total dust emission in Australia (Ginoux et al., 2012; Tegen et al., 2004; Webb and Pierre, 2018). Indeed, disturbed soil and vegetation from land use, such as pastural and agricultural activity in eastern Australia, have caused substantial increase in dust emission and deposition during the 20$^{th}$ century (Brahney et al., 2019; Cattle, 2016). The modulation of land use on dust emission and transport from Australia may also be quantified and compared with natural drivers through future Earth system modeling.

## Data availability

The MODIS Deep Blue aerosol products were acquired from the Level-1 and Atmosphere Archive and Distribution System (LAADS) Distributed Active Archive Center (DAAC), located in the Goddard Space Flight Center in Greenbelt, Maryland (https://ladsweb.nascom.nasa.gov/). The MISR aerosol products were acquired from the NASA Langley Research Center Atmospheric Science Data Center (https://l0dup05.larc.nasa.gov/MISR/cgi-bin/MISR/main.cgi). The AERONET coarse-mode aerosol optical depth data were downloaded from https://aeronet.gsfc.nasa.gov. The NCDC Integrated Surface Hourly Database was accessed from ftp://ftp.ncdc.noaa.gov/pub/data/noaa/. NOAA CPC precipitation data was provided by the NOAA/OAR/ESRL PSD, Boulder, Colorado, USA, from their website at https://psl.noaa.gov/data/gridded/data.cpc.globalprecip.html. NOAA CDR leaf area index was downloaded from

https://data.nodc.noaa.gov/cgi-bin/iso?id=gov.noaa.ncdc:C00898. ESACCI soil moisture data was download from https://www.esa-soilmoisture-cci.org/node/238. The Australian near surface wind speed data was download from https://data.csiro.au/dap/landingpage?pid=csiro%3AWind_Speed.

**Author contributions**

YY conceived the study, analyzed the data and wrote the manuscript with contribution from PG. PG retrieved MODIS DOD data from MODIS Deep Blue aerosol products.

**Competing interests**

The authors declare that they have no conflict of interest.

**Acknowledgements**

This research is supported by NOAA and Princeton University's Cooperative Institute for Climate Science. The authors thank Drs. John Dunne and Khaled Ghannam for their helpful comments on the early version of this paper. We thank the AERONET program for establishing and maintaining the sun photometer sites used in this study. We acknowledge the NCDC for collecting ground observations from global weather stations. We thank the MODIS and MISR teams for providing data and useful discussions.

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

**Table 1 Summary of the observational datasets for environmental variables applied in the current analyses across Australia.**

| Variables | Dataset | Period | Temporal resolution | Spatial resolution | References |
|---|---|---|---|---|---|
| Leaf area index (LAI) | NOAA Climate Date Record (CDR) Leaf Area Index (LAI) and Fraction of Absorbed Photosynthetically Active Radiation (FAPAR) dataset | 2000-2019 | Daily | 0.0833° x 0.0833° | (Vermote and NOAA CDR Program, n.d.) |
| Surface soil moisture | European Space Agency (ESA) Climate Change Initiative (CCI) global satellite-observed soil moisture dataset | 2000-2018 | Daily | 0.25° x 0.25° | (Dorigo et al., 2017) |
| Precipitation | NOAA Climate Prediction Center (CPC) Global Unified Gauge-Based Analysis of Daily Precipitation | 2000-2019 | Daily | 0.5°x0.5° | (Chen et al., 2008) |
| Near-surface wind speed | Commonwealth Scientific and Industrial Research Organisation (CSIRO) Near-surface wind speed dataset | 2000-2018 | Daily | 0.01°x0.01° | (McVicar, 2011) |

**Table 2 ENSO and MJO's modulation on regional DOD during September-February. Analyzed regions include the western Australian (32˚S-24˚S, 110˚E-130˚E), Lake Eyre-Torrens-Frome Basin (32˚S-26˚S, 135˚E-141˚E), and eastern Australia (36˚S-28˚S, 143˚E-150˚E). Analyzed dustiness variables include regional mean DOD and frequency of extreme DOD, namely daily DOD anomaly exceeding three times of interannual standard deviation. The regression analysis is performed with antecedent June Niño 3.4. One, two, and three asterisks indicates statistically significant regression or anomaly, based on Monte Carlo permutation or bootstrap test with p<0.1, 0.05, and 0.01, respectively.**

| Variable | Region | Reg. on Niño 3.4 | Anomaly by MJO phase (% of mean) | | | | Reg. on Niño 3.4 by MJO phase | | | |
|---|---|---|---|---|---|---|---|---|---|---|
| | | | 1-2 | 3-4 | 5-6 | 7-8 | 1-2 | 3-4 | 5-6 | 7-8 |
| DOD | West | 0.0001 | 14.7*** | 4.68** | -14.73*** | -1.5 | -0.001 | 0.001 | 0.0005 | -0.0003 |
| | Lake Eyre | 0.011* | 2.67 | 7.76** | 2.46 | -2.77 | 0.007 | 0.012* | 0.012* | 0.012 |
| | East | 0.027** | 0.09 | 6.42* | 6.53** | -0.12 | 0.019* | 0.029** | 0.034*** | 0.025* |
| Frequency of extreme DOD (%) | West | 0.10* | 16.93*** | 4.56** | -7.63 | 0.15 | -0.22 | 0.37 | 0.17 | -0.14 |
| | Lake Eyre | 0.98* | 1.04 | 8.90* | 3.21 | -1.02 | 0.45 | 1.14 | 1.44* | 1.02 |
| | East | 3.51** | -2.45 | 0.15 | 8.01** | -0.72 | 1.55* | 3.81** | 4.22*** | 2.37* |

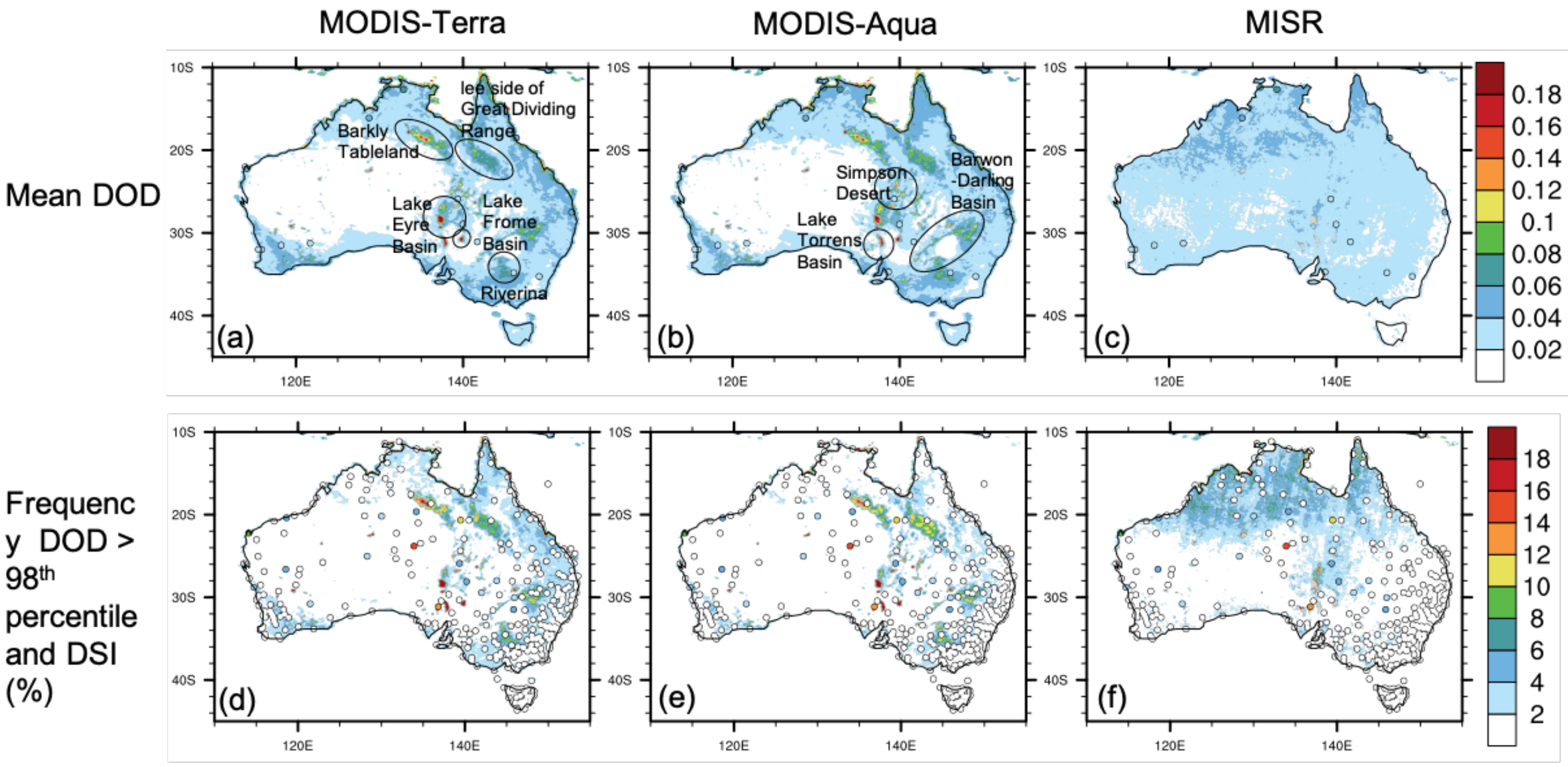

590 **Figure 1: Annual mean dust activity over Australia during 2000-2019. In (a-c), map represents average DOD from (a) MODIS-Terra, (b) MODIS-Aqua, and (c) MISR. Filled circles represent DOD from 18 AERONET sites, identical in (a), (b), and (c). In (d-f), map represents frequency of DOD exceeding the 98th percentile of all observations from each instrument, namely 0.205 for MODIS-Terra, 0.198 for MODIS-Aqua, and 0.096 for MISR. Filled circles represent Dust Storm Index (DSI, %) at 1489 weather stations, identical in (d), (e), and (f). Grey indicates pixels with sampling less than 10 days during the analyzed 20 years.**

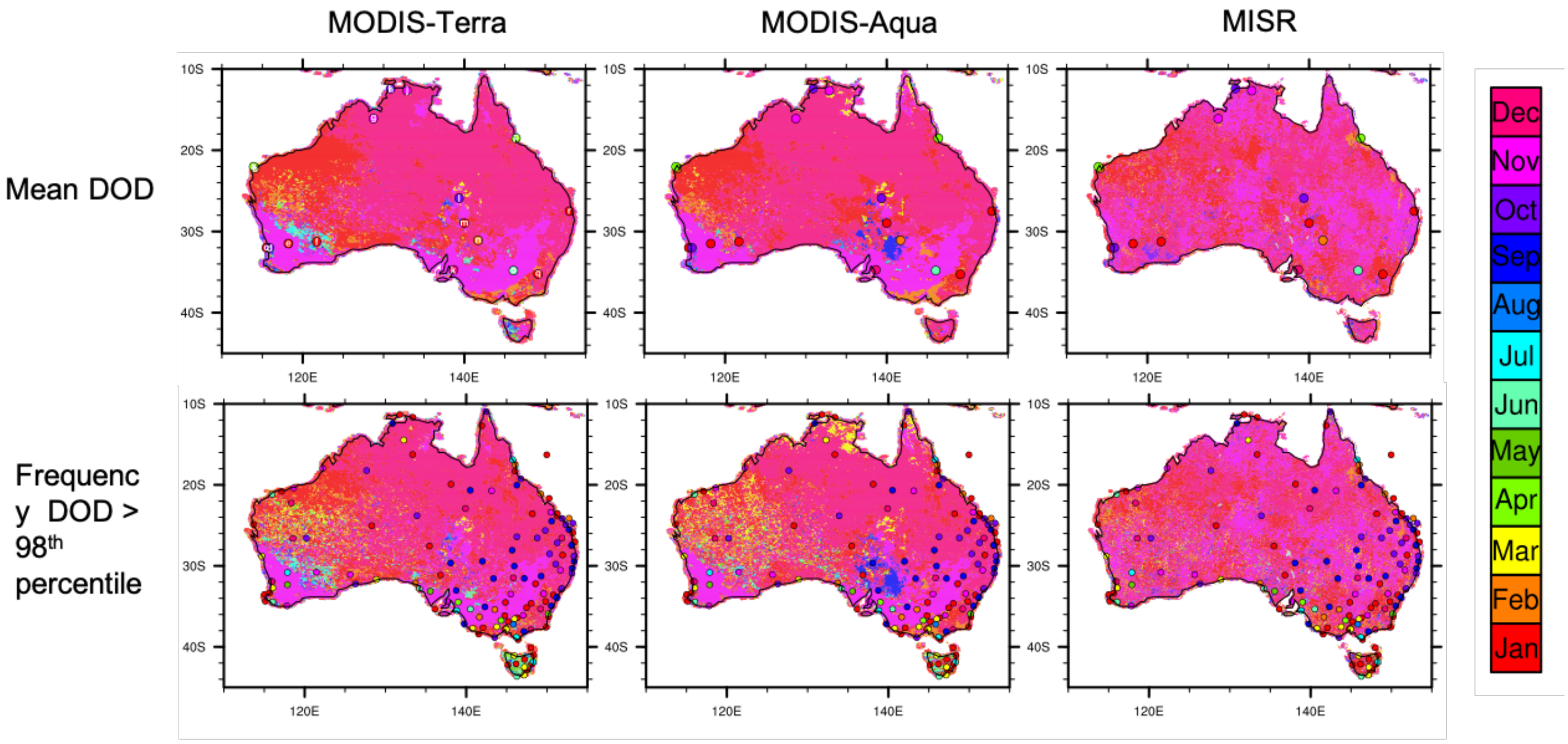

**Figure 2: Month of peak dust activity over Australia during 2000-2019. In (a-c), map represents average DOD from MODIS-Terra, MODIS-Aqua, and MISR. Filled circles represent DOD from 18 AERONET sites. In (d-f), map represents frequency of DOD exceeding the 98th percentile of all observations from each instrument. Filled circles represent DSI from 1489 weather stations.**

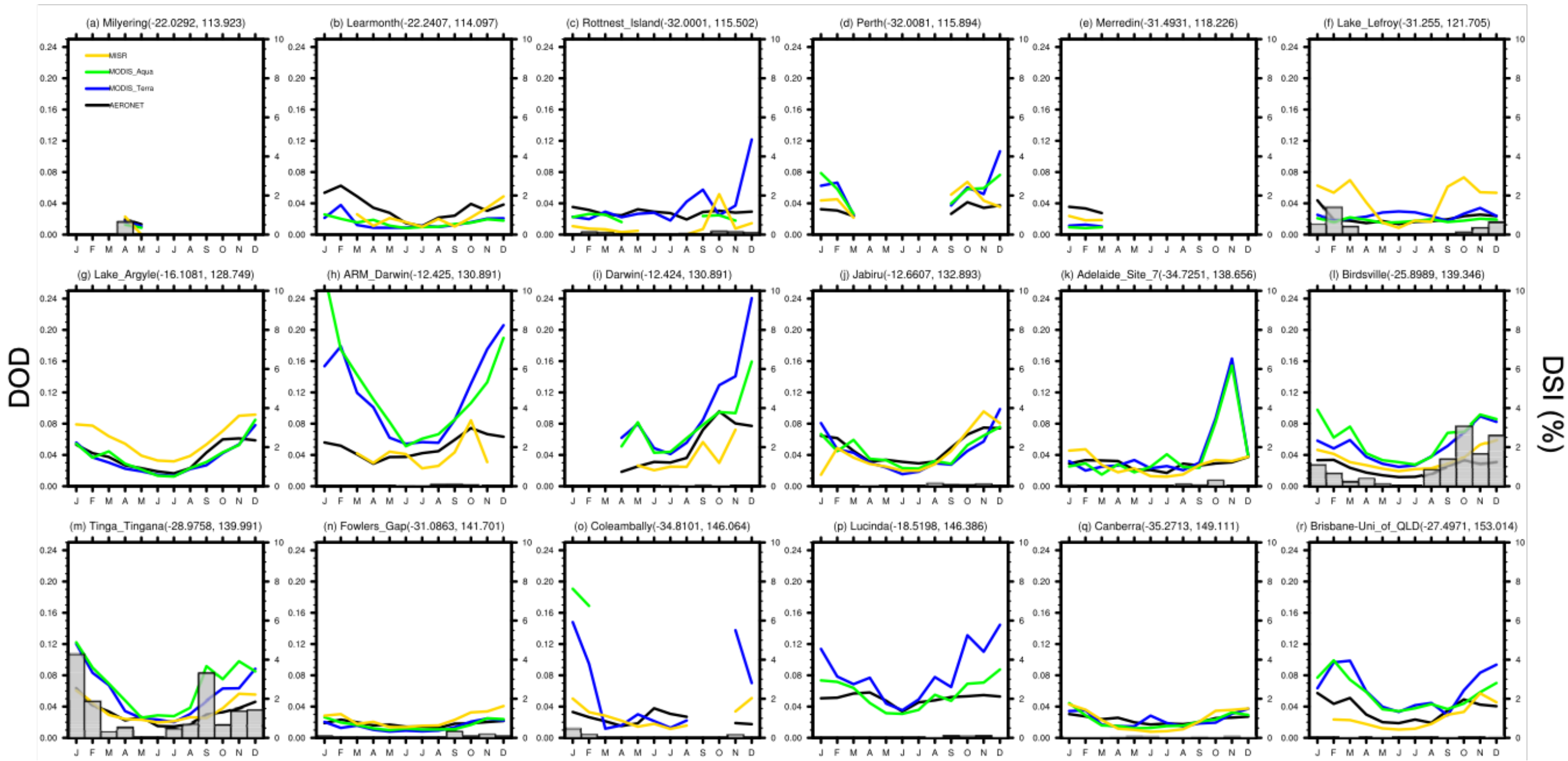

**Figure 3: Seasonal cycle of dust activity at 18 AERONET sites in Australia. Lines represent mean DOD by month from AERONET (black), MODIS-Terra (blue), MODIS-Aqua (green), and MISR (yellow), referring to the left Y-axis. Bars, referring to the right Y-axis, represent mean DSI over weather stations located within 100 km of each AERONET site. The seasonal cycle of dust activity from satellite and in-site observations is obtained from the active years of each AERONET site. The AERONET sites are presented by longitude from west to east, matching the letters in Figure 2a.**

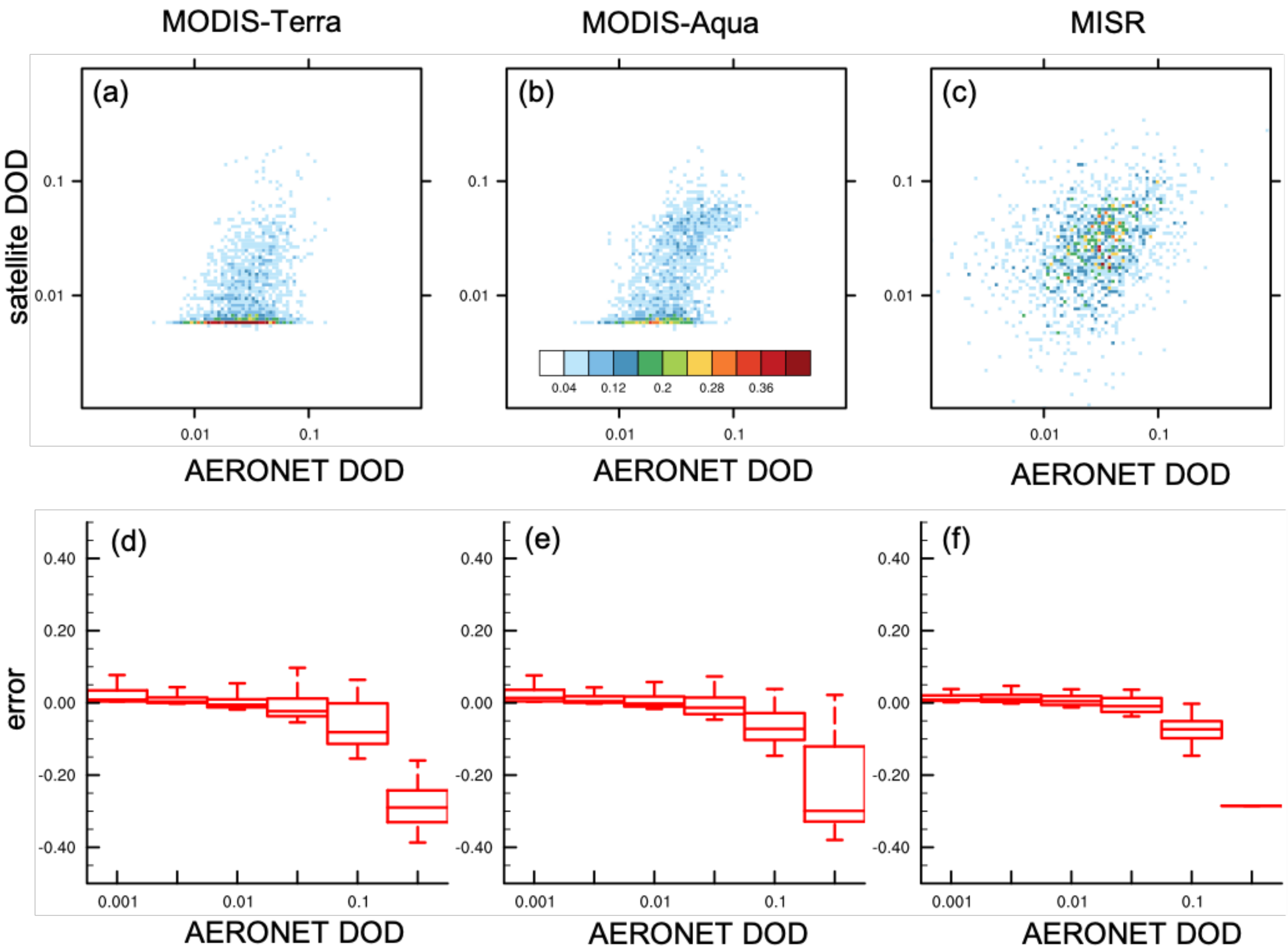

615 **Figure 4: Comparison of satellite DOD against AERONET observation. (a-c) Joint probability density (%) of collocated DOD from AERONET and (a) MODIS-Terra, (b) MODIS-Aqua, and (c) MISR. (d-f) Boxplot of the difference in collocated DOD between (d) MODIS-Terra, (b) MODIS-Aqua, and (c) MISR, and AERONET, as a function of AERONET DOD. The boxplots show the 5$^{th}$, 25$^{th}$, 50$^{th}$, 75$^{th}$, 95$^{th}$ percentiles of the DOD difference.**

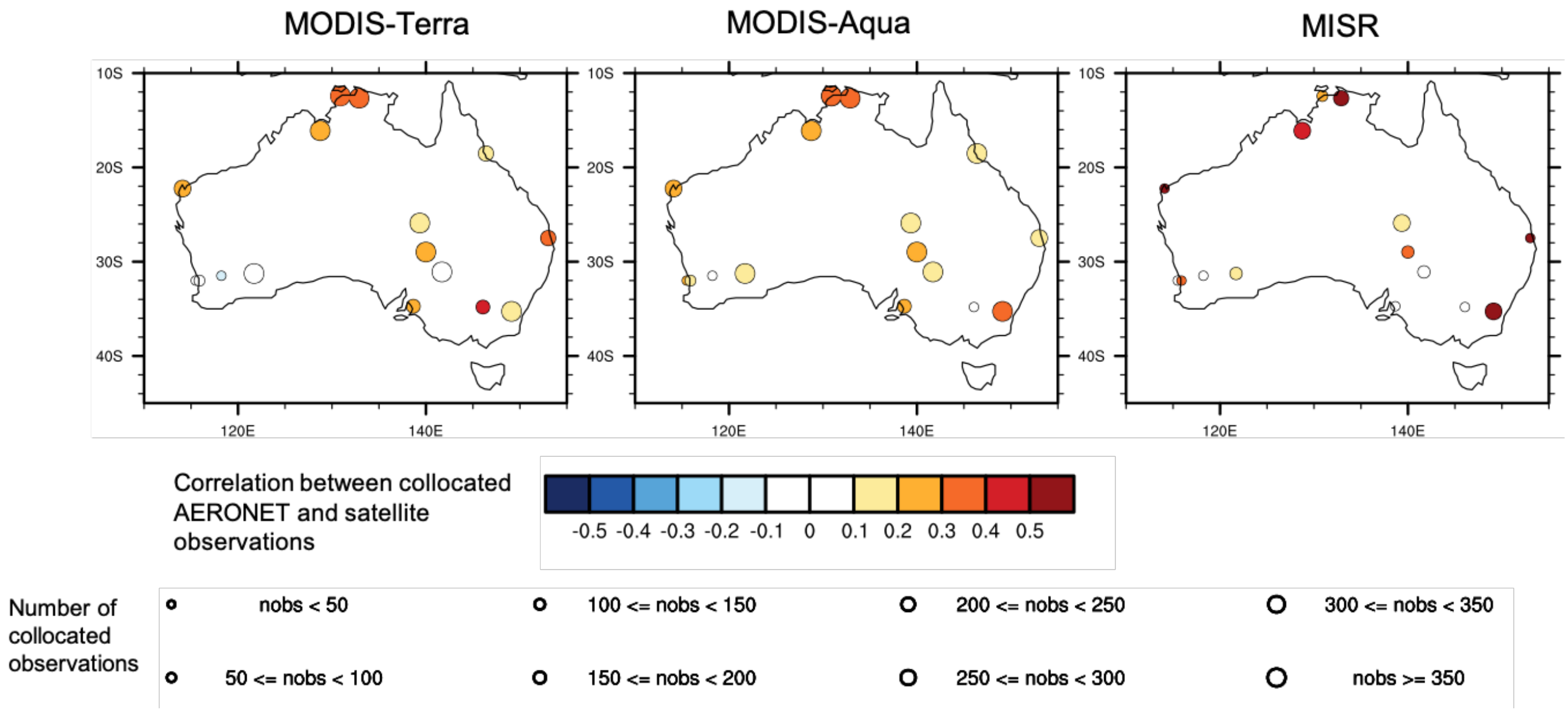

**Figure 5: Temporal correlation between collocated DOD from AERONET and satellite instruments at 18 AERONET sites in Australia. Size of dots indicates the number of collocated observations. A missing circle in (c) indicates no collocation between MISR and AERONET observations.**

630

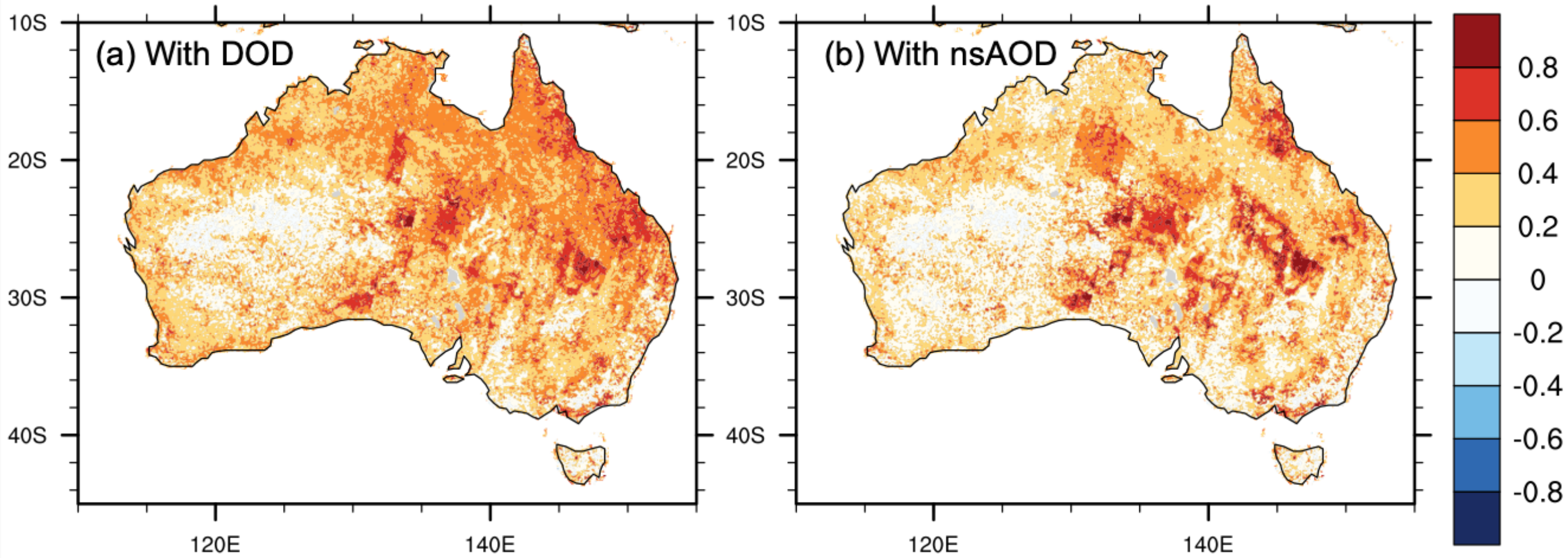

**Figure 6: Temporal correlation between MODIS-Terra DOD and MISR (a) coarse mode AOD (DOD) and (b) nonspherical AOD (nsAOD). Number of collocations per pixel between MODIS and MISR during 2000-2019 generally varies from 100 to 800 for DOD and 20-200 for nsAOD. Grey indicates areas with fewer than 10 collocations per pixel.**

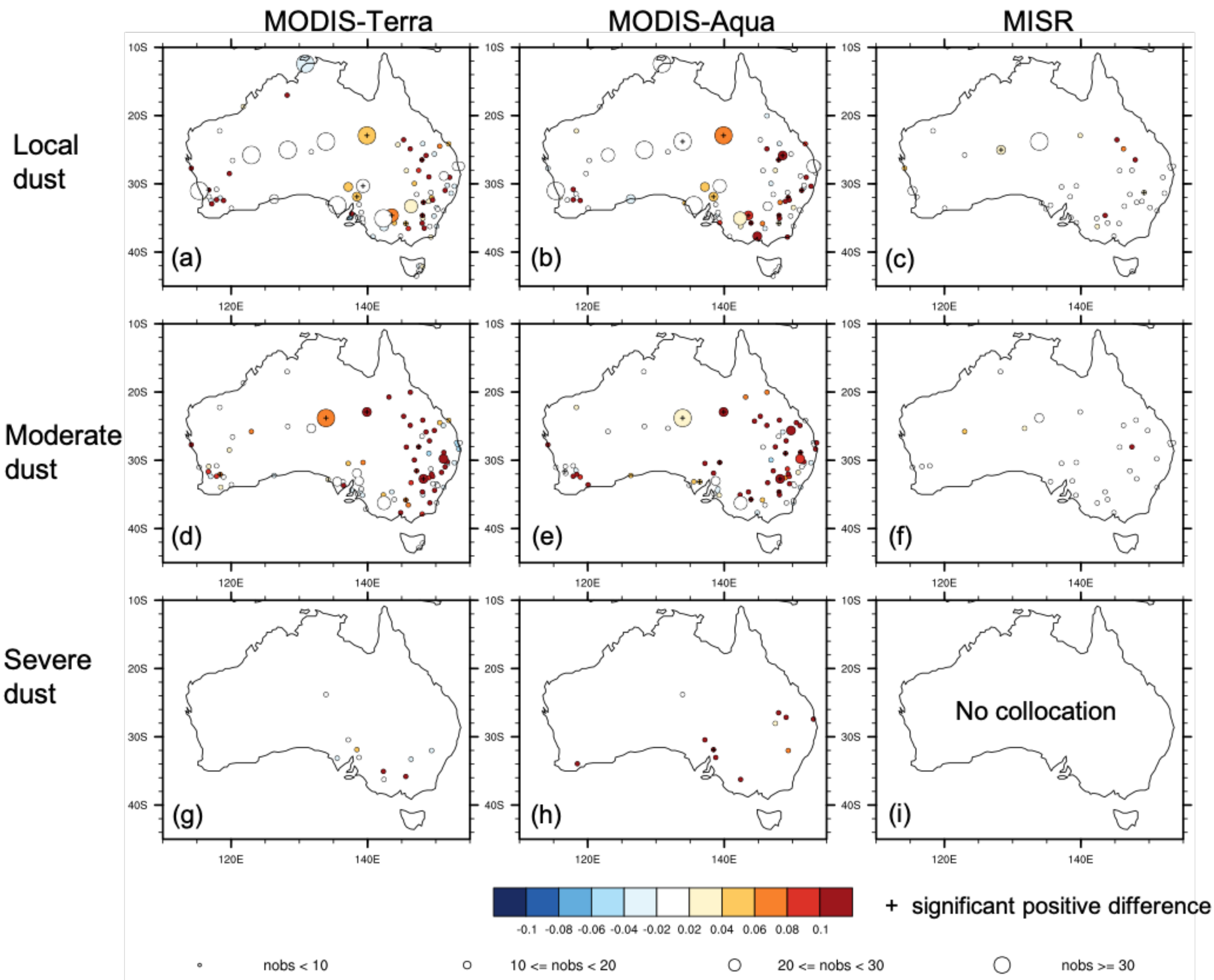

635 **Figure 7: DOD difference between dusty days and clear days at weather stations in Australia. Color of filled dots represents the difference in DOD from (a, d, g) MODIS-Terra, (b, e, h) MODIS-Aqua, and (c, f, i) MISR between days with no reported dust observation and days with reported (a-c) "Local dust", (d-f) "Moderate dust", and (g-i) "Severe dust". Size of dots indicates number of days with both weather observation and valid satellite retrieval within 25 km at each station. A plus sign indicate significant positive difference in DOD between dusty and clear days, based on Monte Carlo bootstrap test ($p<0.05$).**

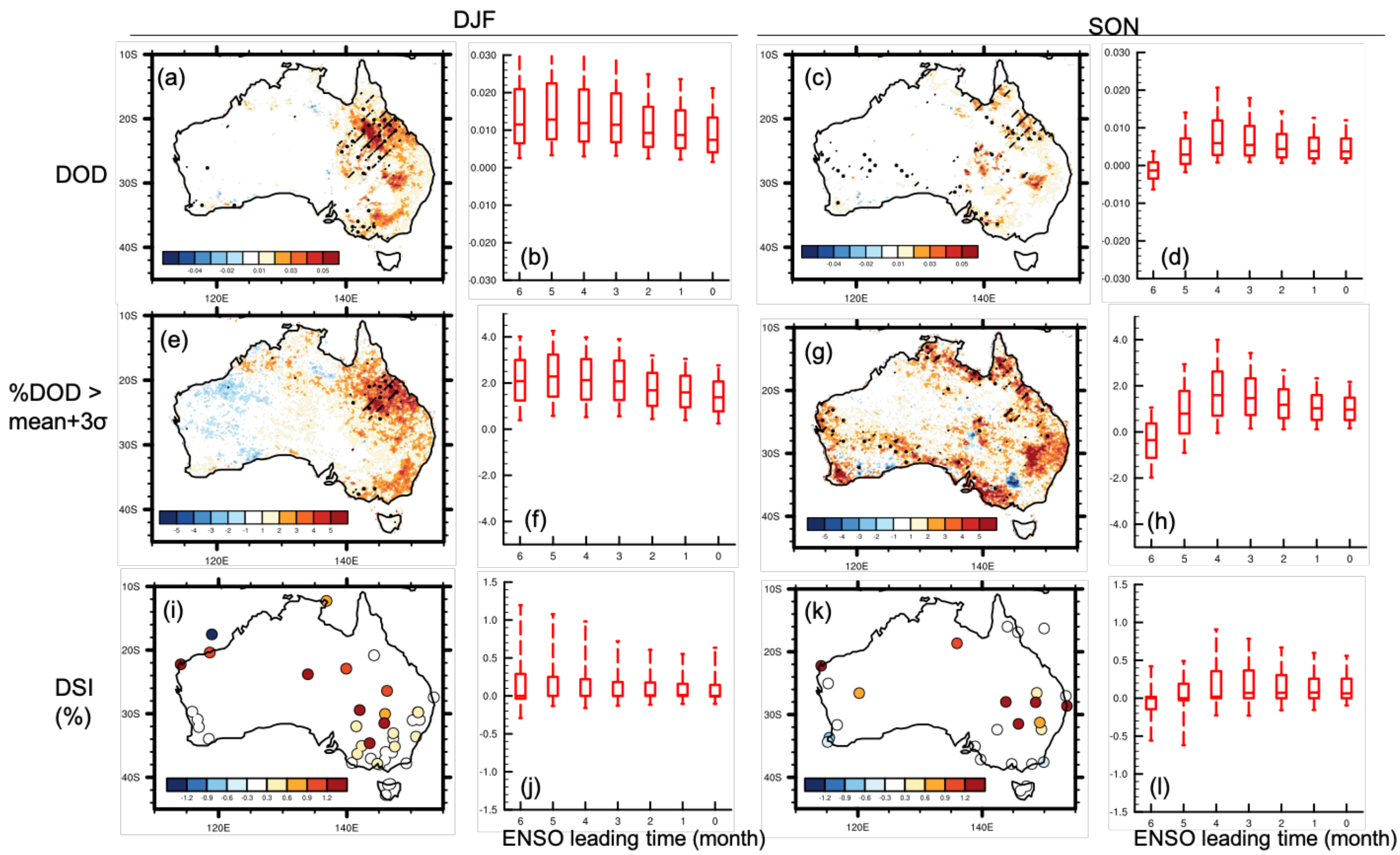

**Figure 8: Regression of anomalies in seasonal dust activity in (a, b, e, f, i, j) December-February (DJF) and (c, d, g, h, k, l) September-November (SON) upon Niño 3.4 at different antecedent time. Analyzed dust variables include seasonal (a-d) DOD averaged from MODIS-Terra and MODIS-Aqua, (e-h) frequency of daily DOD anomaly exceeding three times of interannual standard deviation, and (i-l) DSI. (a, c, e, g, i, k) Regression coefficient between (a, e, i) DJF dust and June-August (JJA) Niño 3.4, and (c, g, k) SON dust and April-June (AMJ) Niño 3.4. In (a, c, e, g), the stitches indicate regions with significant regression coefficient ($p<0.05$), based on Monte Carlo bootstrap test; and the slashes further denotes regions with significant positive correlation between MODIS DOD and MISR nsAOD. In (i, k), only statistically significant regression coefficients ($p<0.05$) are shown. (b, d, f, h, j, l) Boxplot of the regression coefficient of seasonal dustiness on Niño 3.4 at different antecedent time as a function of the leading time of Niño 3.4, showing the 5$^{th}$, 25$^{th}$, 50$^{th}$, 75$^{th}$, and 95$^{th}$ percentiles of regression coefficient at all pixels within the dust source region (25˚S-35˚S, 135˚E-155˚E).**

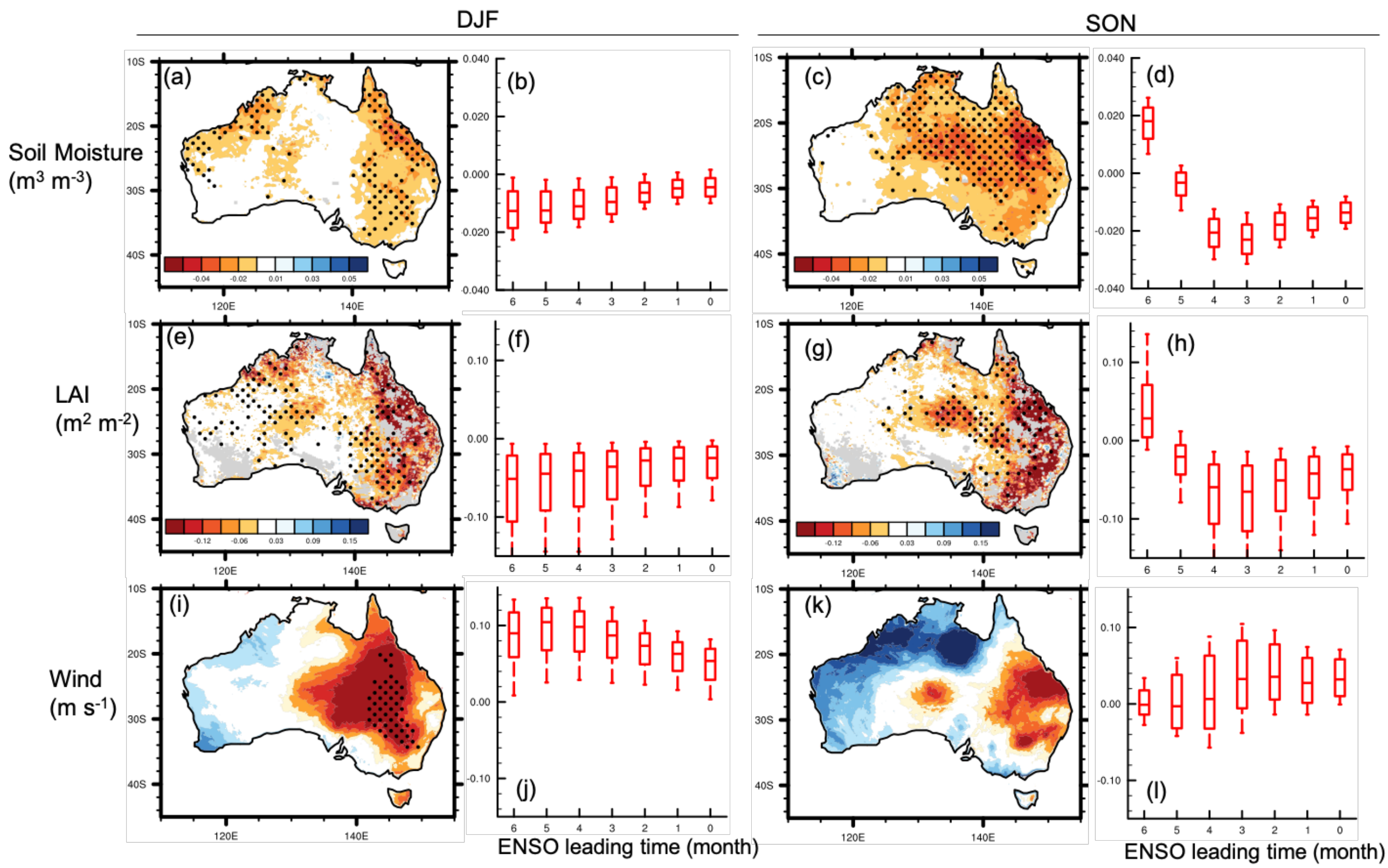

**Figure 9: Regression of anomalies in seasonal LAI, soil moisture, and wind speed in (a, b, e, f, i, j) December-February (DJF) and (c, d, g, h, k, l) September-November (SON) upon Niño 3.4. Figure elements are the same as Figure 8a-h.**

Atmospheric Chemistry and Physics
EGU
Discussions

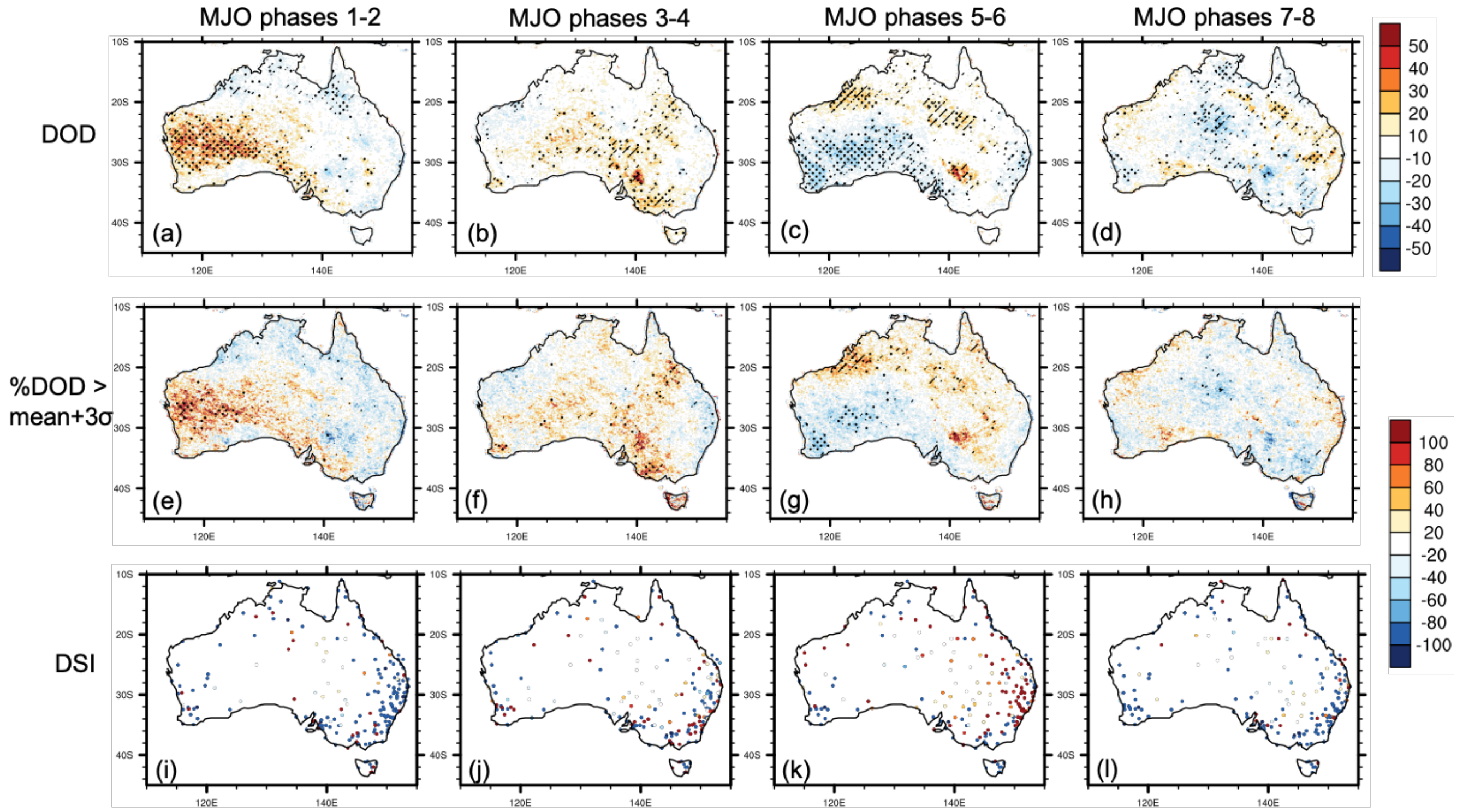

**Figure 10: Mean anomalies (% climatology) in dust activity by MJO phase in September-February. Analyzed dust variables include (a-d) DOD averaged from MODIS-Terra and MODIS-Aqua, (e-h) frequency of daily DOD anomaly exceeding three times of interannual standard deviation, and (i-l) DSI. The anomalies for each consecutive two MJO phases, namely phases 1-2, 3-4, 5-6, and 7-8, are calculated as the percentage differences between these two phases and the long-term average during September-February of 2003-2019. The composites consist of 376, 492, 613, and 450 days with RMM > 1 for phases 1-2, 3-4, 5-6, 7-8, respectively. In (a-h), the stitches indicate regions with significant percentage difference with the climatology (p<0.05), based on Monte Carlo bootstrap test; and the slashes further denotes regions with significant positive correlation between MODIS DOD and MISR nsAOD. In (i-l), only statistical significant percentage differences (p<0.05) are shown.**

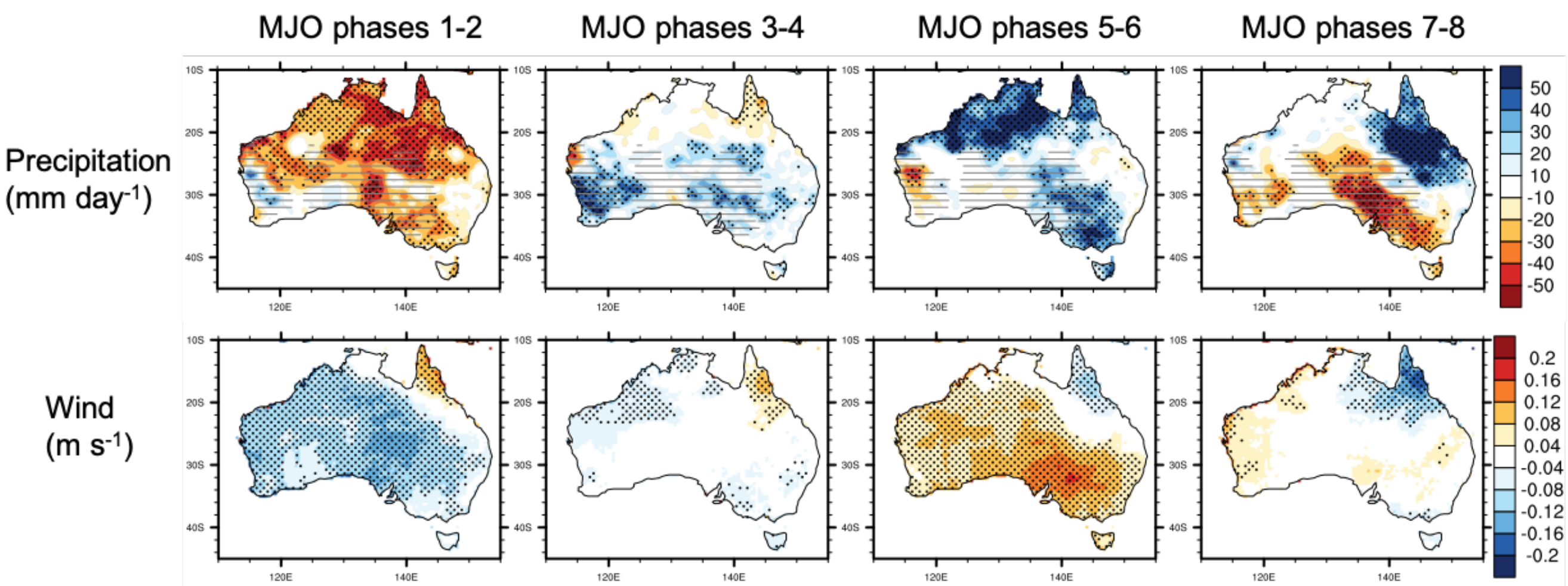

**Figure 11: Mean anomalies in daily precipitation and wind speed by MJO phase in September-February. Figure elements are the same as in Figure 9a-h. Grey dashes indicates areas with seasonal mean rainfall less than 1 mm day$^{-1}$.**

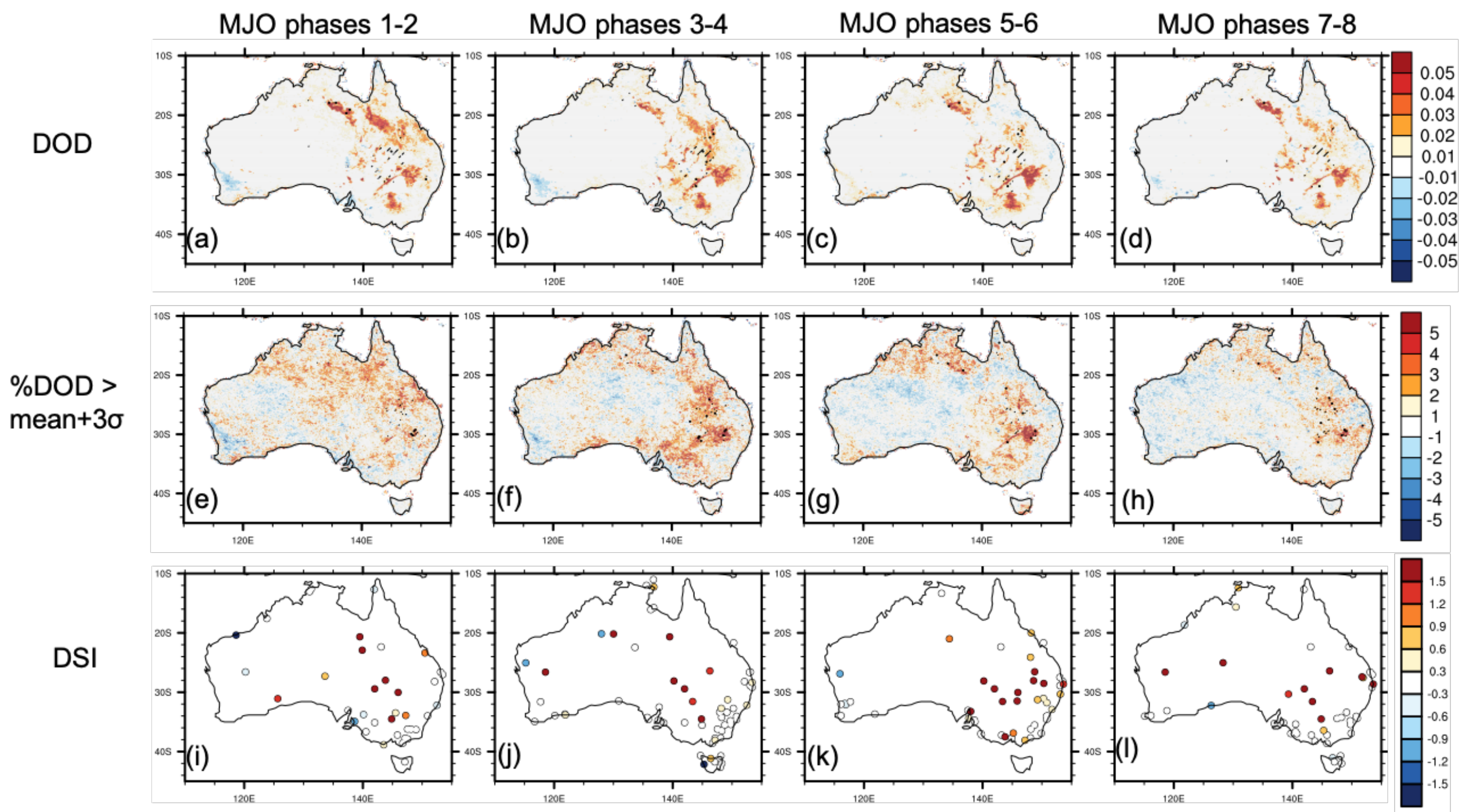

675 **Figure 12: Regression of anomalies in dust activity during each consecutive two MJO phases in September-February upon the antecedent June Niño 3.4 by MJO phase. Analyzed dust variables include (a-d) DOD averaged from MODIS-Terra and MODIS-Aqua, (e-h) frequency of daily DOD anomaly exceeding three times of interannual standard deviation, and (i-l) DSI. The anomalies for each consecutive two MJO phases, namely phases 1-2, 3-4, 5-6, and 7-8, are calculated as the differences between these two phases and the long-term average during September-February of 2003-2019. In (a-h), the stitches indicate regions with significant regression coefficient ($p<0.05$), based on Monte Carlo bootstrap test; and the slashes further denotes regions with significant positive correlation between MODIS DOD and MISR nsAOD. In (i-l), only statistically significant regression coefficients ($p<0.05$) are shown.**