# Peer review of "Assessing the contribution of ENSO and MJO to Australian dust activity based on satellite and ground-based observations"

_Atmospheric Chemistry and Physics, 2020_

## Referee Comment (RC1) · Anonymous Referee #1 · 30 Dec 2020

**Overall comments**

This paper first compares several satellite dust products in Australia against in-situ observations, and then looks at MJO and ENSO - driven variations in dust optical depth. It seems that detailed comparisons of this type have not previously been done. The ENSO-driven dust signatures confirm expectations from previous work, but the MJO ones are novel and look a bit hard to explain (at least to me).

I think this is a useful study but with a number of problems that I think will require major revisions to address.

[Figure]
Interactive comment

1. Many aspects of the data and analysis are confusingly, incompletely or misleadingly described (see examples under Detailed Comments). These problems need to be corrected.

2. The authors have not considered serial autocorrelation in their data and therefore may have overestimated the statistical significance of relationships. How likely this is to be a problem depends on what time averaging interval they used for doing the regression tests (which they don't say). They need to check this.

3. I find the authors too ready to declare success in their satellite data evaluation. In some respects, particularly the climatology and to some extent the seasonal cycles, the satellite and in-situ datasets agree very well. But in terms of day to day (I think) variations they don't look very good. Moreover some comparisons show MISR very different from MODIS (in particular, they look totally different in Fig. 7). And I'm not sure about the MJO or ENSO signatures, which are a better test (day to day variations will be noisy even if the instruments are working well, and climatology is perhaps too easy). The ENSO comparisons (Fig. 8) don't seem to agree well in DJF, with high DSI signatures at several sites where the satellite shows little, and a swath of near-zero DSI signatures in the southeast where the satellite shows dusty conditions. They look better in SON but there is less signal to work with. And the MJO patterns (Fig. 10) seem to be significantly different in quite a few areas between in-situ and satellite, though it is very hard to see because the in-situ points are so small. I think this needs to be better explored—we would expect all of these platforms to deliver similar results on this, wouldn't we? By the time the authors got to the ENSO and MJO signatures they consolidated to just one satellite dataset since up to then they'd agreed well enough, but I think given the seeming disagreement with in-situ on these signals, the authors need to back up and include the separate satellite products in this analysis. They also need to see if they can explain the satellite-in situ discrepancies on the basis of poor sampling in the in-situ data—they seem to perhaps have been too lenient in

including short time series that will give unstable results.

4. In addition the MJO patterns, to my eye, really don't support the claims by the authors that wind speed and rain explain the dust variations. The ENSO patterns could be explained by either wind or rain, but the MJO ones seem to be explained by neither. This is surprising and a bit discouraging but needs to be conveyed clearly by the authors. I wonder if data quality could be an issue?

5. Even if we ignore the serial correlation issue and accept the significance results shown, the authors have not demonstrated that the interactions reported between ENSO and MJO are statistically significant and I doubt that they are. In Table 2 (last four columns) the authors consider only the null hypothesis of $r = 0$ for each phase of MJO. This simply establishes whether a relationship exists. But to establish that this relationship is modulated by MJO (or vice versa), the relevant null hypothesis is that $r$ is *invariant with the phase* of MJO, not that it is zero. Wherever a significant $r$ is reported in the table during at least one MJO phase, there are generally also strong $r$ for the other phases as well. The fact that $r$ exceeds an arbitrary significance threshold during one phase of MJO and not another is NOT a legitimate demonstration of any real change. Likewise, the maps in Fig. 12 don't look that different to me, and the variations among the MJO phases are probably well within statistical sampling uncertainty. Finally the authors don't offer any convincing hypothesis to explain the interactions claimed. They should either do the correct tests to confirm this interaction is real, or remove it from the paper.

**Detailed comments**

Entire manuscript: there are numerous minor grammatical errors that should be fixed by having the manuscript copy-edited. If the journal doesn't do this then the authors should find a way to have it done.

38: this statement is an exaggeration—I doubt dust from one continent is the only thing controlling biological productivity in the Southern Ocean, and it is surely not the only thing (or even a dominant thing) controlling carbon uptake (compared to, say, the ocean circulation).

52: It would be helpful to mention what the difference was (is there some region that the weather-station studies identified as a dust source that was not identified by the others?)

95: this raises a point not yet mentioned, which is what aspects of precipitation do we expect to influence dust? Do extremes matter (I would think not), or is the most important aspect the time lag between rain events (what I would expect)? Are there studies linking these aspects to MJO or ENSO?

Section 2.1: Please say what the time resolution is of the datasets (monthly? Daily?) It is stated under Aeronet that you average monthly data to get annual means but that's the first we've heard of any time resolution and surely your analysis is not all based on annual means? I didn't find any statement of this until the caption of Fig. 11.

103-4: please write out DOD (I assume it is dust optical depth but you never say). I don't think column-integrated extinction is correct (the extinction will be 1–exp(–DOD) right?)

116: can you explain a bit more about how this estimate works? It seems to depend on dust being a different size from non-dust aerosol. From later text I gather this is actually a coarse-mode AOD—tell us!

132: I don't understand this, it does not seem consistent. The AOD in Section 2.1.1 is the MODIS total AOD, no? And the MISR nonspherical AOD is meant to be an estimate of the DOD (this is what you say in the previous sentence)? I am guessing that (1) is designed to give a coarse-mode AOD (which you call there the DOD)? Please explain

this better. And I think it would be clearer to define a (total) AOD; a coarse-mode AOD (perhaps "cmAOD"), which you get from MODIS via (1) and from MISR as a product; and a non-spherical AOD (nsAOD) which you get from MISR. The latter two can be taken as approximations to the true DOD and tested as such.

141: Here again it would be better, rather than confusingly renaming products, to "call a spade a spade" and refer to this as the Aeronet cmAOD. Especially since later (line 210) you acknowledge that it is measuring sea salt in coastal areas (not just dust).

164-174: This random sample re-ordering test will not account for serial autocorrelation in the data, because any that is present will be destroyed in the scrambled synthetic data series. Please check the autocorrelation time scale of the two time series—if at least one of them decorrelates within a couple of time steps you are OK. Otherwise one way to deal with this is to randomly shift, rather than scramble, one of the time series relative to the other, which will preserve any serial autocorrelation. It will limit the number of distinct synthetic samples you can generate but you should have enough. Also, as noted before please indicate what the time resolution is at which you are doing the resampling.

Figure 1: I found it confusing to have different regions shown in panels (a) and (b) of Fig. 1, suggesting some difference in what Terra and Aqua MODIS are seeing. In fact the two appear essentially identical, but this is obscured by the different labels. I suggest maybe generating a separate figure that is just a map showing and identifying all the regions, and then maybe reproducing some of the ellipses from that map (with no labels) in each of the other panels for reference to help match dust blobs to geographic regions. Apart from that, what do unfilled circles mean in the Aeronet data? Do these mean values of less than 2?

Table 2: You don't say what the numbers in the third column mean (I assume p-value), nor what the units are for the "Reg" quantities. The Reg values are very hard to interpret

since they depend on the amplitude of the Nino 3.4 index; it might be more useful to show correlation coefficients.

212-222: Please explain how you get the seasonal cycle peak month. Do you (I hope) fit a sinusoid to the monthly means? Pick the highest month (I hope not)? If there are two similar peaks in different months for example, the latter method could produce unstable results and seemingly large discrepancies may not be statistically or physically significant. I am worried by the fact that the Terra and Aqua datasets sometimes show rather different peak months even though in Fig. 1 they look indistinguishable—this could be because of an unstable method of identifying the peak month, or the lack of a strong seasonal cycle in either dataset (if the seasonal cycle amplitude does not exceed some threshold I'd suggest blanking out any measure of its phase).

217: Surely with only 1 1/3 years of data you should not try to compute a seasonal cycle?!? Please set a minimum number of years and tell us what that is.

Figure 3: First, please use a larger font, this is barely readable even if I zoom in. Second, please identify which quantities from MISR is being plotted (and, why not show both of them?)

225-235: Doesn't Aeronet give a point measurement, which may be a noisy thing to compare to a large satellite footprint? I see no evidence, at least in Fig. 4a and c, that the satellites are saturating systematically at high DOD values. Instead it just looks like the measurements are noisy—there is a lot of scatter at all DOD values and the correlation is rather low. When binned according to one of the two variables (implicitly assuming that one is 'truth') this will always lead to biases at the high and low end as shown in panels (d-f) even if there are no actual biases, because random errors in the bin variable are causing aliasing via systematic binning errors. How are your results affected if you do more temporal averaging of the data before computing the regression? It will likely improve.

Figure 8: Please clarify whether dust is leading or lagging ENSO. Also please clarify what lag is shown in the maps (I assume lag zero but it needs to say). Finally, it is confusing to have the y-axis located at a lag of six, I would expect it to be at zero. I expect people will misread this and think the leftmost bar is the lag-zero one.

275: This story does not seem to match what is in the figures. The wind speeds are indeed higher in MJO 5-6 (Fig. 11), but the dust is no higher than during the other phases (Fig. 10). Moreover the pattern of winds over the four MJO phases if anything seems opposite to that of dust, with the highest dust anomalies (western region during MJO1-2 in particular) coinciding with below-average winds. On the other hand, the ENSO signals (Figs. 8-9) do look as expected.

Figure 9: There is no color bar for panels i,k.

---

## Referee Comment (RC2) · Anonymous Referee #2 · 19 Jan 2021

This study investigates variability of Australian dust and how ENSO and MJO contribute to the dust variability by using dust optical depth proxies from satellite remote sensing measurements (MODIS-Terra, MODIS-Aqua, and MISR) and dust index(DSI) from weather stations. The study includes two parts: (a) inter-comparisons of remote sensing measurements of dust, and (2) regression analysis of MODIS dust optical depth upon Nino index and MJO index. The paper would be a significant contribution to the study of Australian dust (which has been understudied). But authors should fix grammar errors (asking a native speaker of English to proofread the paper or through copy-editing service), clarify data used, and improve quality of figures.

line 29: "surroundings" should be "surrounding". change "aerosol loading to the atmosphere" to "aerosol loading in the atmosphere". line 35-36: awkward sentence. line 38: "largely determine" may be changed to "affect" line 41-42: awkward sentence line 116-117: could you elaborate how MODIS DOD is derived? line 127: people usually use MISR non-spherical AOD to approximate dust optical depth. Here coarse-mode AOD is used instead. Because of MISR's limited spectral range, MISR coarse-mode AOD may have large uncertainties. Could you comment on which one is a better proxy for dust optical depth? line 146-147: better to mention the temporal resolution of AERONET observations. line 167: Will the regression analysis offer causal-effect relationship? line 184-185: don't quite understand this sentence. line 232: "temporal correlations" is confusing. it is simply hourly DOD scatterplot between MODIS and MISR, right? line 251: "at various antecedent time"....For those regression maps, what "antecedent time" is used? Figure 2: denote panels with a, b, c, d, e, and f. How did you get "peak" month if DOD has no statistically significant seasonal variation? In fact, figure 3 shows seasonal variation more clearly. Figure 3: the figure is too small and has bad quality. I would suggest that 1st and 5th panels in top row be removed because these two sites only have 1 or 3 monthly data. Then you will have 16 stations. You can split 16 stations to 4 rows by 4 columns, enlarge the figure. Also try to avoid using "yellow" line. Figure 4: can you provide correlation coefficients? change y-axis "error" to "Satellite-AERONET DOD" Figure 5: how about change "temporal correlation between collocated DOD ...." to "Correlation between collocated hourly DOD from AERONET and satellite measurements..."? "A missing circle in (c) indicates ....." which one is (c)? Figure 6: again, "Temporal correlation" is not easy to understand. Why does MISR nsAOD have less data points thanMISR nsAOD? Figure 8: "Regression of .....at different antecedent time". For (a, c, e, g, f, I), I can understand that 7 different time has been used to calculate the regression. But for those maps (e.g., a, c, e, g, i, k), what antecedent time has been used? Maybe you could consider to split the figure into two, one for line graph and one for map. Figure 9: same comments as in Figure 8. Figure 11: add "surface" before "wind speed".

СЗ

---

## Author Comment (AC1) · 17 Mar 2021

The comment was uploaded in the form of a supplement:
https://acp.copernicus.org/preprints/acp-2020-1206/acp-2020-1206-AC1-
supplement.pdf

---

## Author Response (AR1)

Dear Dr. Balkanski and two reviewers,

Thank you for your valuable comments and suggestions on our manuscript. Corresponding to your suggestions, we made the following changes:
1) Clarifying the data used in the current study;
2) Improving figure quality;
3) Applying data availability screenings to the in situ observations;
4) Redoing seasonal cycle analysis;
5) Modifying statistical significance test;
6) Improving result interpretation; and
7) Correcting grammatical errors.
Responses to each of your comments follow below.

Yan and Paul

**Anonymous Referee #1**
**Overall comments**
This paper first compares several satellite dust products in Australia against in-situ observations, and then looks at MJO and ENSO - driven variations in dust optical depth. It seems that detailed comparisons of this type have not previously been done. The ENSO-driven dust signatures confirm expectations from previous work, but the MJO ones are novel and look a bit hard to explain (at least to me).
I think this is a useful study but with a number of problems that I think will require major revisions to address.

1. Many aspects of the data and analysis are confusingly, incompletely or misleadingly described (see examples under Detailed Comments). These problems need to be corrected.
Thank you for the suggestions on clarifying our data and analysis. We made corrections or clarifications following your detailed comments.

2. The authors have not considered serial autocorrelation in their data and therefore may have overestimated the statistical significance of relationships. How likely this is to be a problem depends on what time averaging interval they used for doing the regression tests (which they don't say). They need to check this.
We clarify the time averaging intervals with the regression analysis in the revised manuscript on lines 192-194, read as "*The influence of ENSO on DOD and DSI are quantified based on regression of seasonal average of daily DOD and occurrence of extremely high daily DOD during December to February (DJF) and September-November (SON) upon antecedent three-month averaged Niño 3.4 (sample size = 17 based on 17 years of data)*".
The autocorrelation of the key dust and environmental variables do not show significant memory, as noted on lines 203-205, read as "*Given the insignificant autocorrelation at a one-year lag with all the dust and environmental variables across the major dusty regions in the central and southeastern Australia (Figure S1), the current statistical significance test does not account for the potential problem with random scrambling caused by autocorrelation*".

3. I find the authors too ready to declare success in their satellite data evaluation. In some respects, particularly the climatology and to some extent the seasonal cycles, the satellite and in-situ datasets agree very well. But in terms of day to day (I think) variations they don't look very good. Moreover some comparisons show MISR very different from MODIS (in particular, they look totally different in Fig. 7). And I'm not sure about the MJO or ENSO signatures, which are a better test (day to day variations will be noisy even if the instruments are working well, and climatology is perhaps too easy). The ENSO comparisons (Fig. 8) don't seem to agree well in DJF, with high DSI signatures at several sites where the satellite shows little, and a swath of nearzero DSI signatures in the southeast where the satellite shows dusty conditions. They look better in SON but there is less signal to work with. And the MJO patterns (Fig. 10) seem to be significantly different in quite a few areas between in-situ and satellite, though it is very hard to see because the in-situ points are so small. I think this needs to be better explored—we would expect all of these platforms to deliver similar results on this, wouldn't we? By the time the authors got to the ENSO and MJO signatures they consolidated to just one satellite dataset since up to then they'd agreed well enough, but I think given the seeming disagreement with in-situ on these signals, the authors need to back up and include the separate satellite products in this analysis. They also need to see if they can explain the satellite-in situ discrepancies on the basis of poor sampling in the in-situ data—they seem to perhaps have been too lenient in including short time series that will give unstable results.

We have intensively revised the figures and texts in Section 3.1. At the end of this section on lines 294-299 of the revised manuscript, we include a revised summary of the comparison, read as "*Overall, the general consistency between MODIS DOD and collocated AERONET cmAOD, MISR cmAOD and MISR nsAOD, and qualitative consistency between MODIS DOD and station dust observations provides confidence in the credibility of MODIS DOD records in the representation of dustiness over the bare ground and sparsely vegetated regions of Australia. Considering the temporal and spatial coverage of each dataset, only results from MODIS DOD, represented by the average between the morning (Terra) and afternoon (Aqua) overpasses, and station DSI are presented in the following section of climatic modulation on Australian dustiness*".

We have applied a data coverage screening to the station data, described on lines 194-196 of the revised manuscript, read as "*The regression analysis is performed with stations that have more than two weeks' daily DSI during the focal season (DJF or SON) of at least 12 out of the 17 years*", and on lines 218-220 of the revised manuscript, read as "*The composite analysis is applied to stations that have more than seven days' daily DSI in each MJO phase group during the dust season (September to February) of at least 12 out of the 17 years*", and on lines 227-228, "*For a specific station in specific MJO phases during the dust season, the phase-specific, seasonal mean DSI is only computed when daily DSI is available on at least seven days, otherwise reported as missing value*". These data screenings lead to a better agreement between satellite DOD and station DSI in Figure 9 (original Figure 8), Figure 11 (original Figure 10), and Figure 13 (original Figure 12). We also enlarge the station points in the figure for better visualization.

Unfortunately, we are not able to repeat the regression and composite analysis with MISR cmAOD or nsAOD data due to MISR's sparse sampling. We briefly discuss this issue on lines 232-239 of the revised manuscript, read as "*Given the single assumption on dust particle shape involved in nsAOD, the MISR nsAOD is often regarded as a better proxy of DOD than coarse-mode AOD. But the limited temporal coverage of MISR makes it less useful for studying the day-to-day variations and extreme events of dust activity, especially corresponding to MJO. Typically, MISR only samples about five days during each MJO phase group (phases 1 – 2, 3 – 4, 5 – 6 and 7 – 8) per dust season (September to February) over most pixels in Australia. Furthermore, the retrieval of the dust-smoke mixtures, typically present over the southeastern shrublands and grasslands in Australia, is subject to huge uncertainty in the operational MISR aerosol product (Garay et al., 2020; Kahn et al., 2010). Therefore, MISR cmAOD and nsAOD are analyzed here only to support the reliability of MODIS DOD in representing dust activity*".

4.  In addition the MJO patterns, to my eye, really don't support the claims by the authors that wind speed and rain explain the dust variations. The ENSO patterns could be explained by either wind or rain, but the MJO ones seem to be explained by neither. This is surprising and a bit discouraging but needs to be conveyed clearly by the authors. I wonder if data quality could be an issue?

In the revised manuscript, we expand the explanation of this surprising wetter – dustier pattern on lines 324-333 of the revised manuscript, read as "*Surprisingly, the enhanced dustiness over the central and eastern Australian dust hotspots seems to be associated with anomalously wet conditions during all MJO*

*phases. Given that central-southern Australia generally receive less than 1 mm of rainfall on an average day, we hypothesize that over these arid or semi-arid regions, enhanced rainfall during the MJO phases 3-6 in austral spring and summer associated with enhanced convection and occurrence of thunderstorms support higher occurrence of haboob type of dust events. Several case studies have reported haboob dust events in the central and eastern Australia (McTainsh et al., 2005; Shao et al., 2007). Strong et al. (2011) found that about 24% of dust storms in the lower Lake Eyre Basin during 2005-2006 are associated with thunderstorms. Our alternative hypothesis relies on the supply of fine particles by occasional flooding from MJO-induced storms. For supply-limited and/or transport-limited dust sources such as those in southeastern Australia, lack of occasional storms under drier conditions usually leads to the failure of sediment replenishment, thereby leading to anomalously inactive dust emission (Arcusa et al., 2020; Bullard and Mctainsh, 2003)".*

We also summarize these hypotheses in the discussion section, on lines 389-392 of the revised manuscript, read as "*One explanatory hypothesis for this relationship builds partly on the occurrence of haboob dust storms and its connection with MJO-induced anomalies in deep convection over the southeastern Australia. An alternative hypothesis relies on the supply of sediments by MJO-induced storms and their resulting occasional flooding. Our results shed light on a potential linkage between extreme precipitation and enhanced dust emission in Australia*".

5. Even if we ignore the serial correlation issue and accept the significance results shown, the authors have not demonstrated that the interactions reported between ENSO and MJO are statistically significant and I doubt that they are. In Table 2 (last four columns) the authors consider only the null hypothesis of r = 0 for each phase of MJO. This simply establishes whether a relationship exists. But to establish that this relationship is modulated by MJO (or vice versa), the relevant null hypothesis is that r is *invariant with the phase* of MJO, not that it is zero. Wherever a significant r is reported in the table during at least one MJO phase, there are generally also strong r for the other phases as well. The fact that r exceeds an arbitrary significance threshold during one phase of MJO and not another is NOT a legitimate demonstration of any real change. Likewise, the maps in Fig. 12 don't look that different to me, and the variations among the MJO phases are probably well within statistical sampling uncertainty. Finally the authors don't offer any convincing hypothesis to explain the interactions claimed. They should either do the correct tests to confirm this interaction is real, or remove it from the paper.

Thank you for the constructive suggestion on testing MJO's modulation of the ENSO-dust relationship. In the revised manuscript, we modify the significance test regarding the MJO phase-specific regression and perform a data availability screening for station DSIs, reflected on lines 225-234 of the revised manuscript, "*Further, regression of dustiness upon ONI is performed for each MJO phase group to evaluate potential role of MJO in modulating ENSO's influence on Australian dustiness. Phase-specific, seasonal mean DOD and DSI are calculated before being regressed on antecedent ONI. For a specific station in specific MJO phases during the dust season, the phase-specific, seasonal mean DSI is only computed when daily DSI is available on at least seven days, otherwise reported as missing value. The statistical significance of MJO's modulation on ENSO-dust relationship is assessed by a Monte Carlo test with 1,000 iterations. In each iteration, daily dustiness measures are randomly sampled from the entire dust season with the same size as a particular group of MJO phases and averaged to obtain a random-phase mean dustiness measure for each year. The time series of these random-phase mean dustiness measures is regressed on the antecedent ONI, resulting in a PDF of the regression coefficients to test if the regression coefficient from the realistic, phase-specific dustiness is lower than the 2.5$^{th}$ or higher than the 97.5$^{th}$ percentile of the PDF*".*

With all these changes, MJO's modulation of ENSO-dust relationship appears more robust and consistent between satellite and station observations, as reflected in the revised Figure 13. Our hypothesis is provided on lines 339-345 of the revised manuscript, read as "*We hypothesize that the enhanced response in dustiness across the southeastern Australia to ENSO during MJO phases 3-6 are attributed to the interplay between MJO-induced anomalies in convection, rainfall, and wind and the ENSO-induced*

*anomalies in soil moisture and vegetation. While the dry soils and diminished vegetation caused by El Niño provide favorable conditions for dust emission (Figure 10), the active convections and elevated occurrence of extreme precipitation during MJO phases 3-6, as well as strengthened surface wind during MJO phases 5-6, likely trigger more dust emission and extreme dust events across southeastern Australia through either haboob type of dust events or additional sediment supply by occasional flooding (Figure 12)".*

**Detailed comments**

Entire manuscript: there are numerous minor grammatical errors that should be fixed by having the manuscript copy-edited. If the journal doesn't do this then the authors should find a way to have it done.
Thank you for the suggestion on grammar. We have carefully checked the entire manuscript and corrected grammatical errors.

38: this statement is an exaggeration—I doubt dust from one continent is the only thing controlling biological productivity in the Southern Ocean, and it is surely not the only thing (or even a dominant thing) controlling carbon uptake (compared to, say, the ocean circulation).
This sentence is changed to "*Since most of the Southern Ocean is iron-limited (Sunda and Huntsman, 1997), the transport and deposition of Australian dust affect its productivity and carbon uptake (Boyd et al., 2004; Gabric et al., 2002)*" on lines 35-37 of the revised manuscript, as also suggested by the other reviewer.

52: It would be helpful to mention what the difference was (is there some region that the weather-station studies identified as a dust source that was not identified by the others?)
This sentence has been changed to "*Ginoux et al (2012) further identified agricultural dust sources in the Murray-Darling Basin in southeastern Australia, including the Victorian Big Desert, Riverina, and the Barwon-Darling Basin, consistent with an earlier satellite-based dust source identification (Prospero et al., 2002), model-based wind erodibility during dry years (Webb et al., 2006); but these agricultural dust sources generated minimal dust storm frequency at nearby weather stations (McTainsh, 1989; McTainsh et al., 1989, 1998, 2007; O'Loingsigh et al., 2014)*" on line 48-52 of the revised manuscript.

95: this raises a point not yet mentioned, which is what aspects of precipitation do we expect to influence dust? Do extremes matter (I would think not), or is the most important aspect the time lag between rain events (what I would expect)? Are there studies linking these aspects to MJO or ENSO?
Past studies on rainfall affecting Australian dust emission have mainly focused on long-term rainfall anomalies driven by ENSO. We emphasize the ENSO-induced persistent rainfall anomaly in driving interannual-to-decadal variations in Australian dust emission, on lines 56-60 of revised manuscript, read as "*Observations and General Circulation Models (GCMs) have shown substantial variability in the occurrence and intensity of dust emissions across Australia on the interannual to decadal time scales, primarily driven by persistent anomaly in rainfall associated with Pacific sea-surface temperature (SST) fluctuations, particularly El Niño-Southern Oscillation (ENSO) events (Bullard and Mctainsh, 2003; Evans et al., 2016; Lamb et al., 2009; Risbey et al., 2009; Strong et al., 2011; Webb et al., 2006)*". There has not been published study linking MJO to Australian dust emission, as introduced on lines 94-96 of the revised manuscript, read as "*Despite MJO's critical influence on the regional climate, its direct or indirect role in modulating dust emission or concentration in Australia has, to our knowledge, never been explicitly investigated in either observations or models*".

We outline the current hypotheses regarding ENSO and MJO's modulation of different aspects of precipitation and the resultant response in Australian dust emission, on lines 99-101 of the revised manuscript, read as "*We further provide hypotheses regarding ENSO and MJO's modulation on Australian dust activity, through ENSO's cumulative influence on vegetation and soil properties and MJO's short-term perturbation on convection and extreme precipitatio*n".

We also point out an implication of the current study on linking dust emission with extreme precipitation, on lines 391-392 of the revised manuscript, read as "*Our results shed light on a potential linkage between extreme precipitation and enhanced dust emission in Australia*".

Section 2.1: Please say what the time resolution is of the datasets (monthly? Daily?) It is stated under Aeronet that you average monthly data to get annual means but that's the first we've heard of any time resolution and surely your analysis is not all based on annual means? I didn't find any statement of this until the caption of Fig. 11.

All the original time resolutions are provided in the revised Section 2. Examples include: "*Following Pu et al. (2020), daily DOD is retrieved from collection 6.1, level 2 MODIS Deep Blue aerosol products (Hsu et al., 2013; Sayer et al., 2013)*" on lines 110-111; "*In the current study, Version 23, Level 2, daily MISR 550-nm coarse-mode AOD (cmAOD) and nonspherical AOD (nsAOD) at 4.4-km resolution (Garay et al., 2020) are compared with MODIS DOD*" on lines 130-131; "*The Version 3, level 2 (cloud screened and quality assured), sub-daily AERONET coarse-mode AOD (cmAOD) at 500 nm obtained from the 18 sun photometers across Australia (Giles et al., 2019) and retrieved by the Spectral Deconvolution Algorithm (SDA) (O'Neill et al., 2003) is analyzed here along with DOD from MODIS and cmAOD MISR*" on lines 140-142; and "*Following O'Loingsigh et al. (2014), the daily Dust Storm Index (DSI) at a specific station is a weighted sum of dust activity*" on lines 159-160 of the revised manuscript.

We also describe the temporal aggregation approach for the regression analysis on lines 192-194 of the revised manuscript, read as "*The influence of ENSO on DOD and DSI are quantified based on regression of seasonal average of daily DOD and occurrence of extremely high daily DOD during December to February (DJF) and September-November (SON) upon antecedent three-month averaged Niño 3.4 (sample size = 17 based on 17 years of data)*", and on lines 256-257, read as "*Phase-specific, seasonal mean DOD and DSI are calculated before being regressed on antecedent ONI*".

103-4: please write out DOD (I assume it is dust optical depth but you never say). I don't think column-integrated extinction is correct (the extinction will be 1–exp(–DOD) right?)

This sentence is changed to "*Dust optical depth (DOD) is a column-integration of extinction coefficient by mineral particles*" on line 105 of the revised manuscript.

116: can you explain a bit more about how this estimate works? It seems to depend on dust being a different size from non-dust aerosol. From later text I gather this is actually a coarse-mode AOD—tell us!

We now explicitly state that dust are generally coarser particles on lines 113-121 of the revised manuscript, read as "*To account for dust's absorption of solar radiation and separate dust from scattering aerosols, such as sea salt, we require the single-scattering albedo at 470 nm to be less than 0.99 for the retrieval of DOD. Based on the size distribution of dust towards the coarse range and to separate it from fine particles, DOD is retrieved as a continuous function of AOD and Ångström exponent:*

*$DOD = AOD \times (0.98 - 0.5089\alpha + 0.051\alpha^2)$.* (1)

*This retrieval of DOD is on the basis of Ångström exponent's sensitivity to particle size, with smaller values of Ångström exponent indicating larger particles (Eck et al., 1999), and the previously established relationship between Ångström exponent and fine-mode AOD (Anderson et al., 2005). In short, MODIS DOD represents the optical depth of absorbing, coarse-mode aerosols that are often dust over bare ground or sparsely vegetated regions*".

132: I don't understand this, it does not seem consistent. The AOD in Section 2.1.1 is the MODIS total AOD, no? And the MISR nonspherical AOD is meant to be an estimate of the DOD (this is what you say in the previous sentence)? I am guessing that (1) is designed to give a coarse-mode AOD (which you call

there the DOD)? Please explain this better. And I think it would be clearer to define a (total) AOD; a coarse-mode AOD (perhaps "cmAOD"), which you get from MODIS via (1) and from MISR as a product; and a non-spherical AOD (nsAOD) which you get from MISR. The latter two can be taken as approximations to the true DOD and tested as such.

141: Here again it would be better, rather than confusingly renaming products, to "call a spade a spade" and refer to this as the Aeronet cmAOD. Especially since later (line 210) you acknowledge that it is measuring sea salt in coastal areas (not just dust).

We change the naming of different proxies of DOD throughout the section of intercomparing difference sources of DOD proxies. Section 2.1 is retitled as "DOD proxies". Following previous studies using MODIS DOD (e.g. Ginoux et al 2012; Pu et al. 2020), we keep MODIS DOD as it is but outline its physical meaning on lines 120-121 of the revised manuscript, read as "*In short, MODIS DOD represents the optical depth of absorbing, coarse-mode aerosols that are often dust over bare ground or sparsely vegetated regions*". The coarse-mode AOD from MISR and AERONET are now defined as cmAOD, and non-spherical AOD from MISR is defined as nsAOD.

164-174: This random sample re-ordering test will not account for serial autocorrelation in the data, because any that is present will be destroyed in the scrambled synthetic data series. Please check the autocorrelation time scale of the two time series—if at least one of them decorrelates within a couple of time steps you are OK. Otherwise one way to deal with this is to randomly shift, rather than scramble, one of the time series relative to the other, which will preserve any serial autocorrelation. It will limit the number of distinct synthetic samples you can generate but you should have enough. Also, as noted before please indicate what the time resolution is at which you are doing the resampling.

We now include the lag-one autocorrelation maps of dust and environmental variables in Figure S1. There is no significant autocorrelation except wind speed in western Australia. We keep the original random scrambling test but clarify the time resolution and temporal aggregation for each analyzed variable in Section 2.4, on lines 203-205, read as "*Given the insignificant autocorrelation at a one-year lag with all the dust and environmental variables across the major dusty regions in the central and southeastern Australia (Figure S1), the current statistical significance test does not account for the potential problem with random scrambling caused by autocorrelation*".

Figure 1: I found it confusing to have different regions shown in panels (a) and (b) of Fig. 1, suggesting some difference in what Terra and Aqua MODIS are seeing. In fact the two appear essentially identical, but this is obscured by the different labels. I suggest maybe generating a separate figure that is just a map showing and identifying all the regions, and then maybe reproducing some of the ellipses from that map (with no labels) in each of the other panels for reference to help match dust blobs to geographic regions. Apart from that, what do unfilled circles mean in the Aeronet data? Do these mean values of less than 2?

We add Figure 1 that outlines the land cover types and key dust sources in the revised manuscript. The text and ellipses are removed from the original Figure 1 (now Figure 2). Thanks for these valuable suggestions. We also add the corresponding plots for nsAOD in the revised Figure 2 and Figure 3 for completeness. In the revised Figure 2e-h, we use dots to represent stations with annual mean DSI smaller than 2% for better visualization.

Table 2: You don't say what the numbers in the third column mean (I assume p-value), nor what the units are for the "Reg" quantities. The Reg values are very hard to interpret since they depend on the amplitude of the Nino 3.4 index; it might be more useful to show correlation coefficients.

We decide to remove Table 2 from the revised manuscript. All the messages in Table 2 are delivered in Figures 9, 11, and 13.

212-222: Please explain how you get the seasonal cycle peak month. Do you (I hope) fit a sinusoid to the monthly means? Pick the highest month (I hope not)? If there are two similar peaks in different months for example, the latter method could produce unstable results and seemingly large discrepancies may not

be statistically or physically significant. I am worried by the fact that the Terra and Aqua datasets sometimes show rather different peak months even though in Fig. 1 they look indistinguishable. This could be because of an unstable method of identifying the peak month, or the lack of a strong seasonal cycle in either dataset (if the seasonal cycle amplitude does not exceed some threshold I'd suggest blanking out any measure of its phase).

Thanks for the constructive suggestion on analyzing seasonal cycle. In the revised manuscript, we adopt the recommended analysis for getting the seasonal cycle peak month. The approach is described in Section 2.3, read as

"*2.3 Seasonal cycle of dustiness*

*To achieve statistically meaningful analysis of the dustiness annual cycle, the peak month of each dustiness measure, namely DOD from MODIS, cmAOD and nsAOD from MISR, cmAOD from AERONET, and DSI from weather stations, is obtained via a two-step approach. First, a sinusoid function of month is fitted for each dustiness measure,*

$$D(i) = \alpha \sin\frac{i\pi}{6} + \beta \cos\frac{i\pi}{6} + \gamma \qquad\qquad (3)$$

*Where i stands for the calendar month (1 for January, 2 for February, ..., and 12 for December). D(i) is the 20-year average dustiness in month i. α, β, and γ are estimated by minimizing the square-error between the predicted and observed D(i)'s (i = 1 to 12).*

*Then the peak month of dustiness is obtained from the predicted dustiness among 12 months. The peak month is regarded statistically meaningful only if (1) the predicted and observed seasonal cycle of dustiness are significantly correlated with correlation exceeding 0.58 (n = 12), based on the Student's t-test at a significance level of 0.05, (2) the root-mean-square-error between the predicted and observed dustiness is below a quarter of the annual mean dustiness, and (3) the amplitude of the predicted dustiness seasonal cycle (maximum minus minimum) exceeds half of the maximum value among 12 months*".

217: Surely with only 1 1/3 years of data you should not try to compute a seasonal cycle?!? Please set a minimum number of years and tell us what that is.

We now exclude short AERONET and weather station data, as described on lines 142-146 of the revised manuscript, "*In the analysis of annual mean and seasonal cycle, AERONET cmAOD monthly data are first screened by removing those months with fewer than five days of records. To calculate annual means, years with less than five months of records are removed. Annual mean and seasonal cycle are only analyzed for 15 AERONET stations with at least five months' data for at least three years*", and on lines 167-169 of the revised manuscript, read as "*Similar with the AERONET data availability screening, annual mean and seasonal cycle are only analyzed for 182 weather stations with at least five months' effective data, namely with at least five days' DSI available during these months, for at least three years during 2000-2019*".

Figure 3: First, please use a larger font, this is barely readable even if I zoom in. Second, please identify which quantities from MISR is being plotted (and, why not show both of them?)

The figure is revised according to both reviewers' suggestion. Both the cmAOD and nsAOD from MISR are included in the revised figure.

225-235: Doesn't Aeronet give a point measurement, which may be a noisy thing to compare to a large satellite footprint? I see no evidence, at least in Fig. 4a and c, that the satellites are saturating

systematically at high DOD values. Instead it just looks like the measurements are noisy. There is a lot of scatter at all DOD values and the correlation is rather low. When binned according to one of the two variables (implicitly assuming that one is 'truth') this will always lead to biases at the high and low end as shown in panels (d-f) even if there are no actual biases, because random errors in the bin variable are causing aliasing via systematic binning errors. How are your results affected if you do more temporal averaging of the data before computing the regression? It will likely improve.

We rewrite the paragraph of comparing satellite DOD or cmAODs with AERONET cmAOD on lines 270-283 of the revised manuscript, read as "*The general comparison between collocated satellite DOD or cmAOD and AERONET cmAOD exhibits reasonable quality of satellite retrievals over the majority of Australia, but wider spreads of DOD from both MODIS-Terra and MODIS-Aqua, and cmAOD from MISR, especially corresponding to collocated high cmAOD from AERONET (Figures 5-6). The wide spread of MISR cmAOD, compared with collocated AERONET cmAOD, is partly attributed to the limited spectral range of MISR. Very few MODIS DOD retrievals reach lower than 0.005, likely due to the numerical limits of retrieving algorithm. Furthermore, both MODIS and MISR display wider spread at higher DOD or cmAOD and an overall underestimation, especially when AERONET DOD exceeds 0.1 (Figure 5). This underestimation of high optical depth has been reported by previous global validations of total AOD from MODIS (Sayer et al., 2019; Wei et al., 2019) and MISR (Garay et al., 2020), as well as MODIS DOD (Pu and Ginoux, 2018b). The underestimation of high DOD potentially leads to the deteriorated correlations between collocated satellite DOD or cmAOD and AERONET DOD over the dustiest region near the Lake Eyre Basin, compared with less dusty regions in Australia (Figure 6). Given the distinct retrieval algorithms involved in the satellite DOD, cmAOD, and AERONET cmAOD, the moderate but significant correlations (p<0.001) between collocated, thousands of satellite DOD or cmAOD and AERONET cmAOD (Figure 5) demonstrate the reliability of MODIS DOD and MISR cmAOD in representing coarse-mode aerosol loads*".

We test the comparison with longer temporal averaging window, but the correlation and root-mean-square-error does not improve. We briefly discuss this on lines 146-154 of the revised manuscript, read as "*Here a "collocated observation" is identified when there is available MODIS DOD or MISR cmAOD over the 0.1˚ grid covering the AERONET site within ±0.5 hour of the corresponding AERONET site observation. Although further spatial smoothing may improve the consistency between AERONET and satellite measurements (Yu et al., 2013), here we keep the fine satellite pixels to evaluate the accuracy of satellite products at their original spatial resolution. At each AERONET site, one satellite observation is often associated with multiple AERONET observations in time. In this case, AERONET observations are temporally averaged, resulting in only one pair of collocated and averaged satellite-AERONET DOD observations for a given collocated incident at each AERONET site. Larger temporal averaging windows, such as ±1 hour, do not improve the consistency between satellite and AERONET measurements, likely due to the fine spatial scale considered in the current study*".

Figure 8: Please clarify whether dust is leading or lagging ENSO. Also please clarify what lag is shown in the maps (I assume lag zero but it needs to say). Finally, it is confusing to have the y-axis located at a lag of six, I would expect it to be at zero. I expect people will misread this and think the leftmost bar is the lag-zero one.

Thanks for all the suggestions on Figure 8. The caption of the new Figure 9 (original Figure 8) is revised, read as "*Figure 9: Regression of anomalies in seasonal dust activity in (a, b, e, f, i, j) December-February (DJF) and (c, d, g, h, k, l) September-November (SON) upon antecedent Niño 3.4. Analyzed dust variables include seasonal (a-d) DOD averaged from MODIS-Terra and MODIS-Aqua, (e-h) frequency of daily DOD anomaly exceeding three times of interannual standard deviation, and (i-l) DSI. (a, c, e, g, i, k) Regression coefficient between (a, e, i) DJF dust and antecedent July – September (JAS) Niño 3.4 (ENSO leading dust for five months), and (c, g, k) SON dust and antecedent May – July (MJJ) Niño 3.4 (ENSO leading dust for four months). …*". We also denote the time of ENSO and dust on top of the regression maps. The x-axis of the boxplots now starts at lag zero.

275: This story does not seem to match what is in the figures. The wind speeds are indeed higher in MJO 5-6 (Fig. 11), but the dust is no higher than during the other phases (Fig. 10). Moreover the pattern of winds over the four MJO phases if anything seems opposite to that of dust, with the highest dust anomalies (western region during MJO1-2 in particular) coinciding with below-average winds. On the other hand, the ENSO signals (Figs. 8-9) do look as expected.

We change the interpretation of Figure 12 (original Figure 11) on lines 321-333 of the revised manuscript, read as "*During MJO phases 5-6, namely the convection-active phases for Australia, the increased surface wind speed over the majority of the continent, especially over the dust hotspots in the Lake Eyre-Torrens-Frome Basin and Riverina, appears responsible for the enhanced dustiness (Figure 12). Surprisingly, the enhanced dustiness over the central and eastern Australian dust hotspots seems to be associated with anomalously wet conditions during all MJO phases. Given that central-southern Australia generally receive less than 1 mm of rainfall on an average day, we hypothesize that over these arid or semi-arid regions, enhanced rainfall during the MJO phases 3-6 in austral spring and summer associated with enhanced convection and occurrence of thunderstorms support higher occurrence of haboob type of dust events. Several case studies have reported haboob dust events in the central and eastern Australia (McTainsh et al., 2005; Shao et al., 2007). Strong et al. (2011) found that about 24% of dust storms in the lower Lake Eyre Basin during 2005-2006 are associated with thunderstorms. Our alternative hypothesis relies on the supply of fine particles by occasional flooding from MJO-induced storms. For supply-limited and/or transport-limited dust sources such as those in southeastern Australia, lack of occasional storms under drier conditions usually leads to the failure of sediment replenishment, thereby leading to anomalously inactive dust emission (Arcusa et al., 2020; Bullard and Mctainsh, 2003)*".

Figure 9: There is no color bar for panels i,k.

The color bars are added to the corresponding panels of the revised Figure 10 (original Figure 9).

**Anonymous Referee #2**
This study investigates variability of Australian dust and how ENSO and MJO contribute to the dust variability by using dust optical depth proxies from satellite remote sensing measurements (MODIS-Terra, MODIS-Aqua, and MISR) and dust index (DSI) from weather stations. The study includes two parts: (a) inter-comparisons of remote sensing measurements of dust, and (2) regression analysis of MODIS dust optical depth upon Nino index and MJO index. The paper would be a significant contribution to the study of Australian dust (which has been understudied). But authors should fix grammar errors (asking a native speaker of English to proofread the paper or through copy-editing service), clarify data used, and improve quality of figures.

Thank you for the positive comments and constructive suggestions. We have incorporated these in the revised manuscript.

line 29: "surroundings" should be "surrounding". change "aerosol loading to the atmosphere" to "aerosol loading in the atmosphere".

These corrections are made on line 28 of the revised manuscript.

line 35-36: awkward sentence.

This sentence is changed to "*The deposition of transported dust over ocean affects ocean biogeochemistry through changes to the iron supply (Gabric et al., 2010; Jickells et al., 2005)*" on lines 32-34 of the revised manuscript.

line 38: "largely determine" may be changed to "affect"

This sentence is changed to "*Since most of the Southern Ocean is iron-limited (Sunda and Huntsman, 1997), the transport and deposition of Australian dust affect its productivity and carbon uptake (Boyd et al., 2004; Gabric et al., 2002)*" on lines 35-37 of the revised manuscript.

line 41-42: awkward sentence

This sentence is changed to "*Therefore, deeper understanding of the spatio-temporal variations in Australian dust emission and their driving mechanisms will have broad implications on the regional and global climate*" on lines 37-39 of the revised manuscript.

line 116- 117: could you elaborate how MODIS DOD is derived?

The derivation of MODIS DOD is expanded on lines 110-121 of the revised manuscript, read as "*Following Pu et al. (2020), daily DOD is retrieved from collection 6.1, level 2 MODIS Deep Blue aerosol products (Hsu et al., 2013; Sayer et al., 2013), including aerosol optical depth (AOD), single-scattering albedo (ω), and the Ångström exponent (α). All the daily variables are first interpolated to a 0.1˚ x 0.1˚ grid using the algorithm described by Ginoux et al. (2010). To account for dust's absorption of solar radiation and separate dust from scattering aerosols, such as sea salt, we require the single-scattering albedo at 470 nm to be less than 0.99 for the retrieval of DOD. Based on the size distribution of dust towards the coarse range and to separate it from fine particles, DOD is retrieved as a continuous function of AOD and Ångström exponent:*

$$DOD = AOD \times (0.98 - 0.5089\alpha + 0.051\alpha^2). \qquad\qquad (1)$$

*This retrieval of DOD is on the basis of Ångström exponent's sensitivity to particle size, with smaller values of Ångström exponent indicating larger particles (Eck et al., 1999), and the previously established relationship between Ångström exponent and fine-mode AOD (Anderson et al., 2005). In short, MODIS DOD represents the optical depth of absorbing, coarse-mode aerosols that are often dust over bare ground or sparsely vegetated regions*".

line 127: people usually use MISR non-spherical AOD to approximate dust optical depth. Here coarse-mode AOD is used instead. Because of MISR's limited spectral range, MISR coarse-mode AOD may have large uncertainties. Could you comment on which one is a better proxy for dust optical depth?

Thank you for the valuable insights on the uncertainty of MISR coarse-mode AOD. We incorporate this source of uncertainty to the interpretation of the comparison with AERONET on lines 272-274 of the revised manuscript, read as "*The wide spread of MISR cmAOD, compared with collocated AERONET cmAOD, is partly attributed to the limited spectral range of MISR*".

We add the following comment on lines 372-379 in the discussion section: "*Given the single assumption on dust particle shape involved in nsAOD, the MISR nsAOD is often regarded as a better proxy of DOD than coarse-mode AOD. But the limited temporal coverage of MISR makes it less useful for studying the day-to-day variations and extreme events of dust activity, especially corresponding to MJO. Typically, MISR only samples about five days during each MJO phase group (phases 1 – 2, 3 – 4, 5 – 6 and 7 – 8) per dust season (September to February) over most pixels in Australia. Furthermore, the retrieval of the dust-smoke mixtures, typically present over the southeastern shrublands and grasslands in Australia, is subject to huge uncertainty in the operational MISR aerosol product (Garay et al., 2020; Kahn et al., 2010). Therefore, MISR cmAOD and nsAOD are analyzed here only to support the reliability of MODIS DOD in representing dust activity*".

line 146-147: better to mention the temporal resolution of AERONET observations.

The opening sentence of section 2.1.3 is changed to "*The Version 3, level 2 (cloud screened and quality assured), sub-daily AERONET coarse-mode AOD (cmAOD) at 500 nm obtained from the 18 sun photometers across Australia (Giles et al., 2019) and retrieved by the Spectral Deconvolution Algorithm (SDA) (O'Neill et al., 2003) is analyzed here along with DOD from MODIS and cmAOD from MISR*" on lines 140-142 of the revised manuscript.

line 167: Will the regression analysis offer causal-effect relationship?

We add a note in the revised manuscript, stating that "*Although regression analysis does not directly infer causality, the resultant identification of covariability between Australian dust and antecedent ENSO state indicates higher likelihood of the later driving the former than the opposite*" on lines 196-197. We also discuss this source of uncertainty in the discussion section on lines 394-398, read as "*Furthermore, the current hypotheses regarding the influence of ENSO and MJO are established upon regression and composite analyses, which do not directly infer causality. Advanced statistical approaches, such as the Stepwise Generalized Equilibrium Feedback Assessment (SGEFA) (Yu et al., 2017, 2018b), will be useful to evaluate the role of large-scale climate modes and local environmental changes in the emission and transport of Australian dust*".

line 184-185: don't quite understand this sentence.

This sentence is split into two, read as "*Composite analysis is conducted for DOD, frequency of extremely high DOD, and DSI in each of the consecutive two MJO phases (phases 1-2, 3-4, 5-6, and 7-8) during the dust season September – February. The results are expressed as the differences between the phase-specific DOD or DSI and the all-phase seasonal averages*" on lines 216-218 of the revised manuscript.

line 232: "temporal correlations" is confusing. it is simply hourly DOD scatterplot between MODIS and MISR, right?

This sentence is changed to "*Given the distinct retrieval algorithms involved in the satellite DOD, cmAOD, and AERONET cmAOD, the moderate but significant correlations (p<0.001) between collocated, thousands of satellite DOD or cmAOD and AERONET cmAOD (Figure 5) demonstrate the reliability of MODIS DOD and MISR cmAOD in representing coarse-mode aerosol loads*" on lines 280-283 of the revised manuscript.

line 251: "at various antecedent time"....For those regression maps, what "antecedent time" is used?
The caption of the new Figure 9 (original Figure 8) is revised, read as "*Regression of anomalies in seasonal dust activity in (a, b, e, f, i, j) December-February (DJF) and (c, d, g, h, k, l) September-November (SON) upon antecedent Niño 3.4. Analyzed dust variables include seasonal (a-d) DOD averaged from MODIS-Terra and MODIS-Aqua, (e-h) frequency of daily DOD anomaly exceeding three times of interannual standard deviation, and (i-l) DSI. (a, c, e, g, i, k) Regression coefficient between (a, e, i) DJF dust and antecedent July – September (JAS) Niño 3.4 (ENSO leading dust for five months), and (c, g, k) SON dust and antecedent May – July (MJJ) Niño 3.4 (ENSO leading dust for four months) …*".
We also denote the time of ENSO and dust on top of the regression maps in the revised Figure 9.

Figure 2: denote panels with a, b, c, d, e, and f. How did you get "peak" month if DOD has no statistically significant seasonal variation? In fact, figure 3 shows seasonal variation more clearly.

All panels in all figures are denoted in the revised manuscript. Per the suggestion by the other reviewers, we now first fit a sinusoid function for the monthly mean DOD and DSI, then decide the peak month of this sinusoid function. If the sinusoid function is not consistent with the original monthly mean values, or the range of the fitted seasonal cycle is not big enough, then the peak month does not show up in Figure 3. The detailed methodology is described in Section 2.3 of the revised manuscript, read as

*"2.3 Seasonal cycle of dustiness*

*To achieve statistically meaningful analysis of the dustiness annual cycle, the peak month of each dustiness measure, namely DOD from MODIS, cmAOD and nsAOD from MISR, cmAOD from AERONET, and DSI from weather stations, is obtained via a two-step approach. First, a sinusoid function of month is fitted for each dustiness measure,*

$$D(i) = \alpha \sin\frac{i\pi}{6} + \beta \cos\frac{i\pi}{6} + \gamma \qquad\qquad (3)$$

*Where i stands for the calendar month (1 for January, 2 for February, …, and 12 for December). D(i) is the 20-year average dustiness in month i. α, β, and γ are estimated by minimizing the square-error between the predicted and observed D(i)'s (i = 1 to 12).*

*Then the peak month of dustiness is obtained from the predicted dustiness among 12 months. The peak month is regarded statistically meaningful only if (1) the predicted and observed seasonal cycle of dustiness are significantly correlated with correlation exceeding 0.58 (n = 12), based on the Student's t-test at a significance level of 0.05, (2) the root-mean-square-error between the predicted and observed dustiness is below a quarter of the annual mean dustiness, and (3) the amplitude of the predicted dustiness seasonal cycle (maximum minus minimum) exceeds half of the maximum value among 12 months*".

Figure 3: the figure is too small and has bad quality. I would suggest that 1st and 5th panels in top row be removed because these two sites only have 1 or 3 monthly data. Then you will have 16 stations. You can split 16 stations to 4 rows by 4 columns, enlarge the figure. Also try to avoid using "yellow" line.
Thanks for the suggestions on Figure 3. In the revised Figure 4 (original Figure 3), we remove the 1st, 5th, and 14th panel, which have insufficient data to assess seasonal cycle. We describe the data screening on lines 142-146 of the revised manuscript, "*In the analysis of annual mean and seasonal cycle, AERONET cmAOD monthly data are first screened by removing those months with fewer than five days of records. To calculate annual means, years with less than five months of records are removed. Annual*

*mean and seasonal cycle are only analyzed for 15 AERONET stations with at least five months' data for at least three years".*

Figure 4: can you provide correlation coefficients? change y-axis "error" to "Satellite-AERONET DOD"
The sample sizes, correlation coefficients, root-mean-square errors are denoted on the corresponding panel of the revised Figure 5 (original Figure 4). The y-axis labels are revised accordingly.

Figure 5: how about change "temporal correlation between collocated DOD ...." to "Correlation between collocated hourly DOD from AERONET and satellite measurements..."? "A missing circle in (c) indicates ....." which one is (c)?
Thanks for the suggestion on the title of Figure 6 (original Figure 5). It is changed to "*Correlation between collocated hourly cmAOD from AERONET and satellite measurements at 18 AERONET sites in Australia*". We also added panel labels to the revised Figure 6 (original Figure 5).

Figure 6: again, "Temporal correlation" is not easy to understand. Why does MISR nsAOD have less data points than MISR cmAOD?
The title of Figure 7 (original Figure 6) is changed to "*Correlation between collocated, daily MODIS-Terra DOD and MISR (a) cmAOD and (b) nsAOD*". For successful retrieval of nsAOD, images from the multiple cameras of MISR should all pass quality check, while cmAOD only requires image from one camera. This could be one reason for the different data availability between MISR nsAOD and cmAOD.

Figure 8: "Regression of .....at different antecedent time". For (a, c, e, g, f, l), I can understand that 7 different time has been used to calculate the regression. But for those maps (e.g., a, c, e, g, i, k), what antecedent time has been used? Maybe you could consider to split the figure into two, one for line graph and one for map.
Figure 9: same comments as in Figure 8.
In the caption of the revised Figure 9 (original Figure 8) and Figure 10 (original Figure 9), we state the lag-time of these maps: "*Figure 9: Regression of anomalies in seasonal dust activity in (a, b, e, f, i, j) December-February (DJF) and (c, d, g, h, k, l) September-November (SON) upon antecedent Niño 3.4. Analyzed dust variables include seasonal (a-d) DOD averaged from MODIS-Terra and MODIS-Aqua, (e-h) frequency of daily DOD anomaly exceeding three times of interannual standard deviation, and (i-l) DSI. (a, c, e, g, i, k) Regression coefficient between (a, e, i) DJF dust and antecedent July – September (JAS) Niño 3.4 (ENSO leading dust for five months), and (c, g, k) SON dust and antecedent May – July (MJJ) Niño 3.4 (ENSO leading dust for four months)*", and "*Figure 10: Regression of anomalies in seasonal LAI, soil moisture, and surface wind speed in (a, b, e, f, i, j) December-February (DJF) and (c, d, g, h, k, l) September-November (SON) upon Niño 3.4. (a, c, e, g, i, k) Regression coefficient between (a) LAI, (e) soil moisture, and (i) surface wind speed during DJF and Niño 3.4 during antecedent July – September (JAS) (ENSO leading environmental conditions for five months), and SON (c) LAI, (g) soil moisture, and (k) surface wind speed during SON and Niño 3.4 during antecedent May – July (MJJ) (ENSO leading environmental conditions for four months)*". We also indicate the time of antecedent Niño 3.4 in the corresponding regression maps of the revised Figures 9 and 10.

Figure 11: add "surface" before "wind speed".
"Wind speed" is changed to "surface wind speed" in the caption of Figure 10 and Figure 12 (original Figure 9 and Figure 11).